# Quantified effect of seawater biogeochemistry on the temperature dependence of sea spray aerosol fluxes

Karine Sellegri[1], Theresa Barthelmeß[2], Jonathan Trueblood[1], Antonia Cristi[3], Evelyn Freney[1], Clémence Rose[1], Neill Barr[3], Mike Harvey†, Karl Safi[4], Stacy Deppeler[3], KarenThompson[4], Wayne Dillon[5], Anja Engel[2] and Cliff Law[3,5]

[1]Université Clermont Auvergne, CNRS, Laboratoire de Météorologie Physique (LaMP) F-63000 Clermont-Ferrand, France
[2] GEOMAR, Helmholtz Centre for Ocean Research Kiel, 24105 Kiel, Germany
[3] National Institute of Water and Atmospheric Research (NIWA), Wellington, New Zealand
[4] National Institute of Water and Atmospheric Research (NIWA), Hamilton, New Zealand
[5] Department of Marine Science, University of Otago, Dunedin, New Zealand
†Deceased

*Correspondence to*: Karine Sellegri (karine.sellegri@uca.fr)

**Abstract.** Future change in sea surface temperature may influence climate via various air-sea feedbacks and pathways. In this study, we investigate the influence of surface seawater biogeochemical composition on the temperature dependence of sea spray number emission fluxes. Dependence of sea spray fluxes was investigated in different water masses (i.e. subantarctic, subtropical and frontal bloom) with contrasting biogeochemical properties across a temperature range from ambient (13-18°C) to 2°C, using seawater circulating in a plunging jet sea spray generator. We observed a significant increase in sea spray total concentration at temperatures below 8 °C. Specifically, at 2 °C, there was an average 4-fold increase compared to the initial concentration at ambient temperatures. This temperature dependence was more pronounced for smaller size sea spray particles (i.e. nucleation and Aitken modes). Moreover, the temperature dependence varied based on the seawater type and its biogeochemical properties. While the sea spray flux at moderate temperatures (8-11°C) was highest in frontal bloom waters, the effect of low temperature on the sea spray flux was highest with subtropical seawaters. The temperature dependence of the sea spray flux was found to be inversely proportional to the abundance of the cyanobacterium *Synechococcus* in seawater. This relationship allows for parameterizing the temperature dependence of sea spray emission fluxes based on *Synechococcus*, which may be utilized in future modelling exercises.

## 1 Introduction

In the open ocean, bursting bubbles generated by breaking waves cause particles to be ejected into the atmosphere, in the form of sea spray aerosols (SSA) that may influence climate either via scattering solar radiation (Schulz et al., 2006; Bates et al., 2006) or by forming cloud droplets (Pierce and Adams, 2006) or ice crystals (Burrows et al. 2013). Recent modelling studies have highlighted large knowledge gaps related to sea spray emissions, particularly with the submicron size and effect of organic

matter (Bian et al., 2019; Regayre et al., 2020). Sea spray emission fluxes mainly depend on wind speed which determines the
presence of breaking waves, with a threshold of 4 m s-1 above which sea spray is emitted to the atmosphere. Another important parameter is surface seawater temperature (SST), as the size of bubbles rising to the surface, as well as physical parameters such as the thickness of the breaking bubble films, depend on surface tension, viscosity and density which are all a function of seawater temperature. Also, organic matter in surface seawater influences the bubble bursting process, as organics may alter sea spray mass, number and size, and so affect sea spray cloud condensation nuclei (CCN) properties (Fuentes et al. 2010;
Sellegri et al. 2021). Number based emission fluxes are especially important for cloud formation as the CCN number is more important than mass for cloud properties. Fossum et al. (2018) calculate that for high wind speeds, sea salt number concentrations may contribute 100% of CCN concentrations at realistic marine boundary layer cloud supersaturations in the Southern Ocean.

A number of sea salt aerosol emission parameterizations have been developed but the resulting size-dependant sea spray fluxes
range over one order of magnitude or more (de Leeuw et al. 2007, Ovadnevaite et al. 2014), particularly for sizes lower than 150 nm or larger than 1 micron (Grythe et al. 2014). This large discrepancy in sea spray emission flux parameterizations may reflect the different methods used to measure fluxes in ambient air, and also the biophysical properties in different studies. For example, although fluxes are always parameterized as a function of wind speed, they are not always harmonized for a given seawater temperature or biological content, and when they do they show contradicting results.

Using in situ and remote sensing (aerosol optical depth) measurements to constrain a global model, Jaeglé et al. (2011) derived a parametrization with sea spray flux increasing with temperature over the full temperature range investigated (0-30°C). Consistent with these observations Ovadnevaite et al. 2014 report a positive relationship between sea spray mass fluxes and SST, derived from observation of in-situ coastal sea spray aerosol concentrations and meteorological parameters. Also, Grythe et al. 2014 generated a similar positive linear relationship between sea spray mass and SST from a large data set of fixed station
and ship-borne aerosol measurements, using sodium and a source receptor approach combined with a model. Lastly, Liu et al. 2021 derive a SST dependence of sea-spray fluxes , expressed as sea spray mass for different wind speed ranges that agree with the previously mentioned trends. In summary, the majority of studies using ambient aerosol mass have identified a positive relationship between SSA and SST.

Laboratory experiments using a plunging-jet sea spray generator provide a means to investigate the temperature dependence
of sea-spray number flux (in contrast to sea spray mass flux) across various ranges of sea-spray size and temperature. Usually, relationships derived from number concentration fluxes measurements in the laboratory, dominated by submicron sea-spray aerosols, differ considerably from those obtained from optical ambient measurements such as those described above. Martensson et al. (2003) provide two different temperature dependences of the sea-spray flux over the SST range 0-25°C: one for submicron sea spray, showing an increase of the flux with decreasing temperature, and one for the supermicron size range
of sea spray with increasing flux of sea spray with SST. Similar opposing behaviour between submicron and supermicron

particles was reported by Bowyer et al. 1990. Salter et al. (2014) also found differences in the sea-spray fluxes when considering small and larger sizes of sea spray, but they additionally showed different dependences either side of a temperature threshold of 10°C, i.e. increasing fluxes with increasing SST above 10°C and decreasing fluxes with increasing SST below this limit. Schwier et al. (2017) found an increase in submicron sea-spray fluxes with Mediterranean seawater at temperatures in the 22-29 °C, confirming the results from Salter et al. (2014). Recently, Christiansen et al. (2019) provided a detailed study on the impact of air entrainment and water temperature. Their work confirmed the physical impact of temperature on synthetic seawater with an increase in sub-micrometer sea-spray aerosol with decreasing temperature below 6-10°C, and an increase of sub-micrometer sea-spray aerosol with increasing temperature above this range.

Temperature has an impact on seawater viscosity, density and surface tension, with all these parameters increasing when temperature decreases and also impacts on the evaporation rate of the bubble film. Thorpe et al., (1992) used a numerical model that showed a significant decrease of the mean bubble concentration with increasing temperature, with a halving for every 10°C for bubbles with radii in the 10-150 μm range. However, Christiansen et al. 2019 reported little change with temperature in air entrainment flows via a plunging jet system over the temperature range -2 to 35°C, and Zabori et al. (2012) did not detect any change with temperature of the bubble size distribution in natural seawater over the range -2 to 10°C.

It appears from the large discrepancies between temperature dependences of the sea-spray flux derived from these different approaches that the factors controlling the SSA fluxes are not fully understood, and so not well conceptualized and represented. In their study, Forestieri et al. (2018) highlight that the seawater temperature sensitivity of SSA produced in plunging jet experiments differ greatly between synthetic and natural seawaters. Unlike experiments using synthetic salt solutions that show a monotonic increase of the sea-spray fluxes with seawater temperature, the dependence of sea-spray fluxes to temperature in natural seawaters showed substantial inconsistencies. The authors hypothesize that biological processes induce variations in the surface-active species in the seawater that drive changes in the sea-spray fluxes.

Surfactants influence the bubble bursting mechanism by decreasing or increasing bubble lifetime and size (Modini et al. 2013, Tyree et al. 2007). The presence of surfactant in seawater can cause a decrease in the average SSA size (Sellegri et al. 2006), and an increase or decrease of the number production flux of particles ejected (Tyree et al. 2007; Zábori, et al. 2012), depending on the type of surfactant. Surfactants in the seawater may originate from phytoplankton exudates in the form of dissolved organic matter (carboxylic acids, lipids, amino acids, carbohydrates, etc.) (Aluwihare and Repeta 1999; Barthelmeß and Engel 2022). Surfactant production has been demonstrated with healthy cells of different phytoplankton groups (Zutic et al, 1981) and also during grazing of heterotrophic flagellates and ciliates on bacteria (Kujawinski et al, 2002). In Sellegri et al. (2021), it was shown that the SSA number production flux was a function of the nanophytoplankton cell abundances in surface seawater, for a given temperature. In the same study, sea-spray production fluxes were related to seawater fatty acid concentrations, and associated with a change in seawater surface tension. A clear relationship between nanophytoplankton abundance, the release of labile organic matter and surface activity was also recently confirmed by Barthelmeß and Engel

(2022). The role that surfactants play in determining seawater surface tension and the resulting bubbles at low temperatures is however unclear and may depend on the nature of the surfactants present. To address these uncertainties we investigate the

temperature dependence of sea-spray aerosols generated from natural seawater of contrasting water masses of the South-Western Pacific Ocean under a temperature gradient equivalent to the 25-yr average summer seawater temperature range of the Southern Ocean (Auger et al. 2021) and relate this to the biogeochemical properties of the surface water.

## 2 Material and Methods

Measurements were performed during the Sea2Cloud voyage that took place in March 2020 east of New Zealand on board the

R/V Tangaroa (Sellegri et al. 2022). The region east of the New Zealand South Island provides an ideal platform for investigating variability in SSA and its relationship with surface ocean biogeochemistry due to the close proximity of different water masses (Law et al, 2018). The Sea2Cloud voyage occupied frontal water from 17/03/2020 11:00-20 /03/2020 17:00, and then subantarctic water (SAW) until 24/03/2020 4:00, then subtropical waters (STW) until 25/03/2020 8:00, and finally mixed-shelf water to 27/03/2020. The Subtropical Front which runs along 43oS-43.5oS separates subtropical and subantarctic water

and is evident year-round in ocean colour images as an area of elevated phytoplankton biomass relative to the low biomass water masses either side.

Sea spray was continuously generated with a plunging jet system, as described in detail in Sellegri et al. (2023) and previously used in Schwier et al. 2015 and 2017, Trueblood et al. 2021, Freney et al. 2021 and Sellegri et al. 2021. The 10 L tank was operated with a 10 cm seawater depth, so jet and film drops did not interact with the tank's top locate 15 cm above the seawater

level. Given the jets total flowrate of 1.2 LPM, this relatively small seawater volume results in low residence time (4 min), so preventing changes in biology or sedimentation of large species that occur in larger chambers (Dall'Osto et al. 2022). The small dimensions of our system also correspond to a short residence time of air in the headspace (12s),  preventing potential gas-phase reactions with lab air. Eight plunging jets were created by flushing seawater through 1 micrometer orifices that were equally spaced along a ¼" stainless steal tube, located at 5 cm below the tank's top in the chamber diagonal. Jets penetrate the

seawater volume at a depth of 7 cm, and therefore do not interact with the chamber bottom. Free floating bubbles could occasionally meet the tank's wall as they floated away from the center of the tank. For this reason and others such as the continuous jets vs intermittent wave breaking process, fluxes derived from our experiments, similarly to all controlled lab experiments, are necessarily different from the ones obtained from the natural wave breaking in the open ocean. Natural conditions were however mimicked as much as was possible. The sea-spray generation system was operated at near constant

jet flow rate, and reproduces sea-spray size distributions of similar shape to those reported for other plunging jet devices (Fuentes et al. 2010). The system was fed continuously with seawater sampled from a depth of ~6 m by the underway seawater supply. Once a day, the flow-through seawater system of the plunging jet system was switched from seawater directly coming from the ship underway seawater system to cooled seawater that had been stored in a 50 L temperature-controlled reservoir

immediately before the temperature experiments start (filling time less than 5 min, Figure 1). Temperature gradients between
130 2°C and 15°C were applied to the seawater over approximately 1 hour, with an initial decreasing temperature ramp, followed
by an increasing temperature ramp. The experiments were stopped when the temperature-controlled reservoir had emptied.
These relatively fast temperature changes were applied so the seawater biology did not have the time to evolve, our goal being
to investigate the physical dependance of fluxes to instant biogeochemistry. Temperature experiments were performed every
morning around 11 am from 18/03/20 to 26/03/20, with an additional experiment in the afternoon of 26/03/20.

For submicron particles, SSA were taken through a ¼ inch stainless steel line to a 1-m long silica gel diffusion drier followed
by an impactor with PM1 diameter cutoff. Particle size distributions were monitored by a differential mobility particle sizer
system (DMPS) at 1 LPM, with a separate line to a condensation particle counter (MAGIC CPC, flowrate 0.3 LPM) connected
in parallel for validation. The DMPS system was preceded by a soft X-ray aerosol neutralizer (TSI Model 3088) and consisted
of a TSI-type custom-built differential mobility analyzer (length 44 cm) operated at a sheath flow rate of 5.0 L/min for selecting
particle sizee range of 10-500 nm across 26 size bins during a 13 min 40s scan and a TSI CPC model 3010. Relative humidity
at the inlet was monitored, and kept below 35% at all times. Another short, smooth curvature antistatic Teflon ½ inch line
brought the generated SSA to a Waveband Integrated Bioaerosol Sensor (WIBS) for diameters ranging from 500 nm up to
4500 nm.

The flux of SSA was calculated from the SSA total number concentration, as follows:

$$F_{tot} \ (\# \ m^{-2}s^{-1}) = \frac{CN_{tot}*Q_{flush}}{S_{tank}} \tag{1}$$

where $CN_{tot}$ is the concentration of SSA measured from the MAGIC CPC, $Q_{flush}$ is the flushing air flowrate inside the tank's
headspace, and $S_{tank}$ is the surface of seawater inside the tank. In Sellegri et al. (2021), hereafter referred to as SELL21, the
concentration of > 100 nm particles was used as a proxy for CCN concentration. For comparison to SELL21 we also calculated
fluxes of SSA larger than 100 nm. The flux of $CN_{100}$ ($F_{CN100}$) was calculated in a similar manner to Equation (1):

$$F_{CN100} \ (\# \ m^{-2}s^{-1}) = \frac{CN_{100}*Q_{flush}}{S_{tank}} \tag{2}$$

where $CN_{100}$ is the concentration of SSA with a diameter larger than 100 nm. Calibration experiments performed following
the procedure of Salter et al. (2014), enabled to established that the air entrainment flowrate ($F_{ent}$) in our system is 4.5 $L_{air}$ min$^{-1}$
under the jet operational condition (seawater flowrate of 1.25 L min$^{-1}$, orifices' diameters, jet distance to seawater surface).
The set-up used to measure ($F_{ent}$) reproduced one of the 8 plunging jets set in a separate, larger tank, with the same distance to
seawater and seawater depth than the main experimental set-up. For the air entrainment measurements, the jet was enclosed in
a ½" vertical plunging tubing (at 1 cm depth) connected to a TSI flowmeter. The seawater flowrate was varied from 150 to

400 ml min$^{-1}$ and the relationship between seawater flowrate and entrainment air flowrate was fitted to obtain a calibration curve of our set-up. Air entrainment flowrate calibrations were performed at moderate temperatures around 20 °C and also at lower temperatures that showed undetectable influence of the seawater temperature on the air entrainment flowrate.

Given that, according to Long et al. (2011), the flux of air entrained ($F_{ent}$) during wave breaking can be related to a wind speed at 10 m ($U_{10}$) following:

$$F_{ent} = 2 * 10^{-8} U_{10}^{3,74} \tag{3}$$

we calculate that our plunging jet system simulated a bubble volume distribution equivalent to that produced at a wind speed of 9 m s$^{-1}$. For the data acquired with a seawater flowrate that deviated from 1.25 L min$^{-1}$, fluxes were normalized to the 9 m s$^{-1}$ equivalent windspeed with the following relationship:

$$F_{normalized} = F_{original} * \frac{1.25^{2.4}}{Q_{SW}^{2.4}} \tag{4}$$

Where $Q_{SW}$ is the seawater flowrate. Equation (4) was obtained by varying $Q_{SW}$ over a short period (less than an hour) and fitting the flux dependence to $Q_{SW}$. Normalization resulted in less than 30% change in the fluxes for 80% of the data. The seawater surface tension was measured from the ship-board underway seawater line at 08:00, 12:00, 16:00 and 20:00 NZDT using the bubble lifetime method with a Dynotester along a temperature gradient from 2 to 15°C. The temperature gradient for surface tension measurements was achieved on board the ship on fresh seawater samples by first freezing 25 ml seawater sampled in Falcon tubes, with surface tension measured while the sample slowly warmed to ambient temperature; this took less than one hour which limited the time for any seawater biogeochemistry changes to occur. A bias may exist in the surface tension measured here after samples have been frozen, compared to the surface tension of a sample that would have not experienced freezing, due to the impact of freezing on, for example, the rupture of phytoplankton cells releasing organic matter. Future studies should investigate how freezing may impact surface tension.

For the analysis of the seawater biogeochemical properties, discrete water samples were collected from the ship's underway seawater line into an acid-cleaned 10L plastic carboy at 4 hr intervals (00:00, 04:00, 08:00, 12:00, 16:00, 20:00 NZDT). Sample bottles were either processed immediately or stored in the dark in ENGEL portable fridge/freezers units at in situ temperature for the water mass (max. operating temperature: 9°C) and processed within 8 hrs of collection. Sample volumes for filtering were determined from the Ecotriplet fluorescence data noted during sample collection. Seawater aliquots were taken from the

carboy for chlorophyll a (Chl-a), macronutrients, Total Organic Carbon (TOC) and microbial community composition (flow cytometry and microscopy), all of which were analyzed on land post-voyage.

For chlorophyll-a analysis, 250 ml of seawater was filtered through 25 mm GF/F filters and the pigment retained on the filters was extracted with 90% acetone and measured by spectrofluorometry using a Varian Cary spectrofluorometer. For macronutrient analysis, 250 ml of seawater was filtered through 25 mm GF/F filters into 250ml plastic bottles, and nutrients (Ammonia, Nitrate and Nitrite, Dissolved Reactive Phosphorus, and Silica) measured on a SEAL AA3 Autoanalyser (Law et al. 2011). The TOC content of samples was analyzed by catalytic oxidation (TOC-VCSH analyser, Shimadzu) after the
modified protocol of Sugimura and Suzuki (1988) (Engel and Galgani, 2016). Samples were filled into 20 ml pre-combusted glass ampules (8 h at 500°C), acidified with 20 µl 32% HCl, subsequently sealed and stored at 4°C until the analysis.

For Flow-cytometry, duplicated 1.5 mL seawater samples were preserved with a solution containing  0.5% glutaraldehyde (Naik, S.M. and Anil, 2017), flash frozen and stored at -80°C. Samples were analyzed with a BD FACSCalibur instrument and Synechococcus and picoeukaryote cells quantified using TrucountTM beads (Becton Dickinson, Mountain 108 View, CA), as
described in Hall and Safi (2001). Total numbers of heterotrophic bacteria, eukaryotic nanophytoplankton (2-20µm) and prokaryotic and eukaryotic picophytoplankton (<2 µm) were determined by flow cytometry using a BD Accuri™ C6 Plus instrument (BD Biosciences). Bacteria samples were stained with Sybr Green II and a minimum of 20,000 bacteria events were analysed using SSC vs FL1 plot. For the phytoplankton analyses 250 uL of sample was analysed and the eukaryotic plankton populations were identified using a SSC vs FL3 plot while the prokaryotic picoplankton (Synechococcus sp.)
population was identified using a FL1 vs FL2. For microscopy analysis, 500 ml of seawater was fixed to 1% with Lugol's iodine solution and phytoplankton community composition and cell numbers for species >5 µm were determined using optical microscopy, as described in Safi et al. (2007) and references therein.

## 3 Results and discussion

### 3.1.     General feature of the seawaters investigated

The subtropical water north of the Front is warmer and saltier, relative to the colder, fresher, subantarctic water, with lower dissolved inorganic nutrients, particularly nitrogen and TOC. This variation in physical properties and nutrient availability results in different biological communities in the three water masses, with the Front characterized by blooms of different phytoplankton groups in spring and summer (Delizo et al., 2007; Law et al, 2018). Frontal waters contained higher cell abundances of nanophytoplankton (2-20 µm) than any other water masses sampled during the voyage, whereas cell abundances
of the picoplankton (<2 um) *Synechococcus* were conversely higher in subantarctic and mixed/shelf water relative to frontal water (Figure 2). Diatom abundance was correlated with nanophytoplankton cell abundances ($R^2$=0,57, p<0.01). Differences

in biological communities subsequently influence biogeochemical properties of these water masses resulting in differential effects upon aerosol precursors (Law et al, 2018; Sellegri et al, 2023; Rocco et al. 2023).

### 3.2. Sea-spray fluxes at moderate ambient temperatures

The SSA flux measured at ambient SST (13-18 °C) for the different seawater types are reported in Figure 2d. We observed that the SSA flux decreased from biologically-rich frontal waters to subantarctic seawaters that contained lower phytoplankton abundance, reflected by the nanophytoplankton cell abundance (NanoPhyto) (Fig. 2b and Sellegri et al. 2023). In Sellegri et al. (2021), a relationship was found between the flux of SSA >100 nm in size with the NanoPhyto. Recently, Dall'Osto et al. (2022) also found increased SSA number production fluxes from nanophytoplankton-rich seawater but did not quantify the

SSA flux-to-nanophytoplankton cell abundance relationship.

Figure 3 presents this relationship between $F_{CN100}$ and NanoPhyto for the Sea2Cloud data in comparison with datasets from other regions (SELL21). NanoPhyto during Sea2Cloud were higher on average and spanned a larger range (1490±732 cells mL$^{-1}$) than in oligotrophic Mediterranean waters (546±148 cells mL-1) and springtime Arctic waters in which NanoPhyto were almost absent (4.2±4.3 cells mL-1), but were lower than in a nutrient-enriched mesocosm experiment in New Zealand coastal

waters (4880±2390 cells mL-1). When combining these five datasets, the $F_{CN100}$ showed a weaker dependence than reported in SELL21 but still confirmed the linear dependence of $F_{CN100}$ on NanoPhyto, with a significant correlation ($R^2 = 0.78$, $p<0.05$). The coefficients of the revised fit differ only slightly to SELL21 with a greater statistical significance than for the individual datasets ($R^2 = 0.78$ vs 0.31 for the Sea2Cloud dataset). Following the SELL21 approach, the relationship was reformulated with $F_{CCN}$ expressed as a function of both NanoPhyto and $F_{CN100-inorg}$ with $F_{CN100-inorg}$ corresponding to $F_{CCN}$ in the absence of

biological activity (i.e. solely inorganic chemical components), and hence shown as the intercept of the y axis ($2.41 \times 10^5$ m$^{-2}$ s$^{-1}$):

$$F_{CN100} = F_{CN100-inorg} \times (1 + 9.7\ 10^{-3} \times NanoPhyto) \tag{5}$$

The primary motivation for this reformulation was to allow a simpler implementation of the parameterisation in models that already include the calculation of $F_{CCN-inorg}$. As a reminder, the fluxes used to derive Eq. (5) were normalized to $U_{10} = 9$ m s$^{-1}$

and SST=15°C. We cannot exclude that the dependence of $F_{CCN}$ on NanoPhyto changes for different wind regimes, but not having this information we assume here that Eq. (1) applies to the calculation of $F_{CCN}$ regardless of $U_{10}$.

Surface tension of the seawater show large differences between water types for a given temperature at moderate temperatures >8 °C (Figure 2d). Given that both salinity and temperature influence surface tension, we performed a sensitivity test on the potential impact of salinity on the differences in surface tension observed for different seawater types. Salinity ranges of the

240 different seawater types were 34.2-34.4 g L-1 in SAW, 34.4-34.8 g L-1 in frontal and mixed seawaters, 34.8-35.3 g L-1 in STW and 34.4-34.8 g L-1 in mixed seawaters. We calculate that these salinity ranges correspond to to ideal surface tension

ranges of seawater at 15°C (Nayar et al. 2014) of 74.500-74.502 nN m-1 in SAW, 74.502-74.514 nN m-1 in frontal and mixed seawaters and 74.514-74.523 nN m-1 in STW. Consequently, there is negligeable impact on surface tension within the range of salinities observed. We observe that surface tension is highest for the biologically rich frontal seawaters, and lowest for the STW. This result is somewhat different from the results of Poulain and Barouiba (2018), who suggest that the presence of heterotrophic bacteria and their released surfactants induce the prolongation of lifetime (and therefore lower surface tension). Barthelmeß and Engel (2022) reported a dependence of surfactant release to nanophytoplankton abundance in the Baltic Sea.

Figure 2d also shows that surface tension of the different seawater types at 15°C mostly follow the SSA fluxes at ambient temperature close to 15°C. In our system, increased surface tension corroborates higher SSA number fluxes. Lower surface tension induces higher bubble lifetime and therefore thinner bubble films due to increased drainage and increased evaporation. Upon natural bubble bursting (i.e. if no external force is applied on the bubble film), thinner bubble films may generate a higher number of smaller droplets (Lhuissier and Villermaux, 2012; Poulain and Bourouiba, 2018). In our system, bubbles films are prematurely broken by external forces. These external forces are (1) flushing air flow at the surface of the tank seawater surface (similar to the action of wind) (2) seawater jets continuously splashing over the newly formed bubbles, and (3) other bubbles reaching the surface where preexisting bubbles are present. Therefore bubbles in our system likely do not reach a critical thickness such as the ones described in Lhuissier and Villermaux 2012 or Poulain and Bourouiba, (2018).

### 3.3. Temperature dependence of total spray emission fluxes

For all low temperature gradient experiments we observed an increase in sea-spray total number fluxes with decreasing temperature in all seawater types (Figure 4a). There was no or little change in sea spray flux with an increasing temperature ramp above 8-10 °C but a sharp, negative correlation of sea spray flux with increasing temp below that threshold. This feature was previously observed by Salter et al. (2014) with inorganic seawater, and also Hultin et al. (2011) using Baltic seawater and a sea spray generation system similar to this study, although Hultin et al. (2011) reported a SSA emission flux increase with decreasing temperature already below 12°C. We now discuss the potential factors that may be at the origin of this SSA flux temperature dependance.

Our experiments were performed with ambient air temperature and therefore relatively constant air temperature and RH of the incoming flushing air. Hence there is likely a change in the evaporation rate of the bubble film when the SST is decreased. However, the lower the SST in comparison to the air temperature above, the lower the evaporation of the film would be. Moreover, film thinning due to evaporation becomes more important relative to film thinning due to drainage only for very thin films (Miguet et al. 2021). In our system it is likely that bubbles films are broken by external forces before they reach these very thin films at which evaporation matters (below 1 micrometer for millimetric bubbles, achieved after a lifetime of several 10s of seconds, Miguet et al. 2021).

Results of surface tension measurements as a function of sample temperature during unfreezing are shown in Figure 4b. Surface tension also showed a clear increase with decreasing temperature, therefore with higher surface tension corroborating higher SSA fluxes. However, the temperature dependence of surface tension is relatively monotonous (Figure 4b) while the temperature dependence of the SSA flux is exponential (Figure 4a). Also, the slope of surface tension to temperature does not differ from one sample to the other. This is expected as the Eötvös' equation states that the temperature dependance of the surface tension is the same for almost all liquids. SST-dependance of the evaporation rates should also be the same for all samples. One relevant variable that has varied with temperature in relation to the chemical composition of the solution is viscosity. Viscosity sensitivity to temperature depends not only on the concentration of organics but also on the ionic strength of the solution (pH, salinity), and it also increases exponentially with decreasing temperature (Mallet et al. 2020). An increase of viscosity implies an increase of the characteristic viscous time which leads to the decrease of the bubble film cap thinning rate (drainage) (Miguet et al. 2021). Bubble average lifetimes were found to be very sensitive to viscosity, especially when impurities are present (Miguet et al. 2021). Again, our results would suggest that thicker bubble film caps lead to higher concentrations of sea spray when bubble are mainly broken by external forces. Observed differences in thermal behaviours between seawater types would possibly be explained by differences in the sensitivity of different organic's viscosity to temperature.

In order to decouple the biological impact on SSA fluxes at temperatures in the 8-15 °C range from the influence of lower temperature, we normalized the sea-spray flux at a given temperature ($F_T$) with the sea-spray flux at 8 °C ($F_8$). As presented in Figure 4c, this enabled evaluation of the relative increase of the corresponding sea-spray flux with respect to $F_8$ for a given temperature in the same experiment. The normalization also facilitates comparison of the flux-temperature relationship with those observed in the literature.

The best fit for the normalized sea-spray flux temperature dependence is a polynomial fit of the second order of the following form:

$$\frac{F_T}{F_8} = p1 * T^2 + p2 * T + p3 \qquad (2)$$

When fitting each daily temperature experiment individually, we obtain a time series of coefficients $p_1$ $p_2$ and, as $p_1$ and $p_2$ are highly correlated to $p_3$, it can be expressed as a function of $p_3$ in a final relationship:

$$\frac{F_T}{F_8} = p3 + (0,0382 - 0,243p3) * T + (0,0138p3 - 0,02) * T^2$$

(3)

When the fit is performed on all data together, $p_3$=4.343. Therefore, on average, for a temperature of 1 °C, the sea-spray flux is increased by a factor of approximately 4 relative to the flux at 8 °C. This parameterization is only valid for temperatures below 8 °C whereas fluxes can be considered independent of temperature above 15°C.

305 The different sensitivities to temperature of the SSA flux of different seawater types is obviously due to their different biogeochemical properties. The temporal variability of the fitting parameter $p_3$ of equation (3) can be studied as a function of the seawater biogeochemical properties. We searched for correlations between $p_3$ and the different phytoplankton communities (including nanophytoplankton, flagellates, diatoms, and dinoflagellates), bacteria or biogeochemical variables (TOC, amino acids and carbohydrates). No relationship was found except a significant anticorrelation with *Synechococcus* cell abundance
($R^2$=0,72, n=10, p< 0.00001), with the following relationship showing the temperature dependence of SSA as a function of *Synechococcus* cell abundances expressed in cells ml$^{-1}$:

$p_3$=6.54-2.10$^{-5}$*Synechococcus* (4)

The reason for this anticorrelation is not clear. *Synechococcus* spp. is an autotrophic prokaryote (bacterium) which is ubiquitously present throughout oceanic regimes (Zwirglemaier et al., 2007; Six et al., 2021). Synechococcus spp. occur at a
315 wide temperature range from 0 to 30°C but favor conditions around 10°C on a global scale (Flombaum et al., 2013). While low temperature can induce stress in *Synechococcus spp.* acclimated at higher temperatures, differences occurring in metabolite production can be only expected over the course of several hours (therefore over longer times than those of our experiments), while lowered temperatures are hypothesized to immediately slow down metabolic rates (Guyet et al., 2020). We therefore expect a relatively stable concentration and composition of organic matter and cell abundance during the temperature ramp
experiments. As a consequence, the effect of lowered temperature on SSA fluxes is due to a physical impact of temperature on properties of chemical component already present in the seawater at the beginning of each temperature experiment.

*Synechococcus*-derived dissolved organic matter (SOM) is released via secretion, natural cell death, viral lysis, and predation (Jiao et al., 2010; Fiore et al., 2015; Xiao et al., 2020) and contributes to the marine dissolved organic matter pool (Jiao et al., 2011; Gontikaki et al., 2013). The presence of *Synechococcus* spp. was reported to increase organic carbon content in SSA
particles relative to artificial seawater by a factor of 34 (Moore et al., 2010). Rich in nitrate, the largest proportion of SOM is labile and quickly consumed by heterotrophic bacteria, which release exoenzymes to cleave biopolymers (Christie-Oleza et

al., 2015; Zheng et al., 2021). Surfactants released by bacteria may facilitate substrate degradation, however, also replenish the pool of surfactants present in seawater (Sekelsky and Shreve, 1999).

The anticorrelation with *Synechococcus* that we observe could indicate a release of SOM during their decay arising from
mortality, viral lysis or predation, rather than secretion (which would be positively related to *Synechococcus* cell abundance). Surfactant release was also anticorrelated to *Synechococcus* abundance in the Baltic Sea, and was interpreted as grazing of nanophytoplankton on *Synechococcus* cells (Barthelmeß and Engel, 2022). In theory, we could thus explain the enhanced SSA flux within the lower temperature regimes by (a) SOM released at the expense of *Synechococcus* cell abundance (release during cell destruction, or b) SOM secreted by *Synechococcus* with higher viscosity than other organic matter types since viscosity
prevents drainage and results in larger bubble film thickness and (c) bacterial release of exoenzymes/ surfactants following the destruction of *Synechococcus spp.* cells to facilitate SOM degradation and d) the products of this process i.e. that the surfactant pool is replenished due to cleaved biopolymers from heterotrophic prokaryotes enzymatic activity. Bacterial abundance does not represent their metabolic activity or secretion rate, as bacterial cells can also remain inactive (Lebaron et al., 2001), and so a lack of correlation with bacterial abundance does not necessarily falsify hypothesis (c) and (d). However, we cannot exclude
that the observed anticorrelation with *Synechococcus* is coincidental rather than causal and clearly more work is needed to test these different hypotheses. Next, we investigate if seawater biogeochemistry also influences the size of SSA emissions at low temperatures.

### 3.4.     Size segregated sea-spray fluxes

The aerosol size distributions in the 10-4000 nm range, from merged SMPS and WIBS size distributions, were normalized to
the total sea-spray concentrations in order to investigate changes in size distribution shape rather than number (studied section 2.3). These were averaged over two different temperatures ranges, 2-3 °C and 7-9 °, and fitted with a combination of single lognormal modes (Figures S1 and S2). In the moderate temperature range (7-9°C), we found 4 modes in the submicron range and two modes in the supermicron range that best characterize the sea-spray aerosol, with characteristics summarized in Table 1. These characteristics are very similar to that reported for sea spray generated with the same device using Mediterranean
water (Schwier et al. 2017, Sellegri et al. 2021), and also to sea-spray size distributions generated with other jet-based approaches that showed a dominant mode centred around 100 nm (Sellegri et al. 2006, Fuentes et al. 2010; Christiansen et al. 2019). Christiansen et al. 2019 report that the shape of the particle size distributions and mode contributions are only slightly affected by addition of algae. We also find that the variability of the SSA size distributions at moderate temperature is low between samples, as also pointed out by Sellegri et al. (2021) for seawater from various geographical origins.

When comparing size distributions in the two temperature ranges (Figure 5), we find that the shape of the size distribution is not preserved at 2-3 °C relative to 7-9 °C. We observe an average 15% decrease of the modal diameters at the low temperatures compared to moderate temperatures which is consistent for all modes (Table 1). Also, the fraction of the smaller particle sizes

(nucleation and Aitken modes) relative to the total sea-spray concentration increases at cold temperatures compared to warmer
ones, whereas it decreases for the largest sizes. The average ratio of number concentrations of SSA smaller than 50 nm to the total SSA concentration was $0.19 \pm 0.03$ in the 7-9 °C range, while it was $0.35 \pm 0.04$ in the 2-3 °C range, indicating that the size distribution variation due to low temperature significantly exceeded the variability of size distributions due to seawater types at moderate temperature.

In order to quantify changes of sea-spray concentrations per size range, we calculated modal concentrations by summing particle concentrations in particle size bins within each mode: nucleation mode (11 nm - 15 nm), Aitken mode (20 nm - 44 nm), accumulation mode 1 (68 nm - 142 nm ), accumulation mode 2 (267 nm - 430 nm) and coarse mode (710 nm - 4485 nm). The relation to temperature of each modal concentration normalized to its concentration at 8-10 °C was then plotted for each individual mode and linearly fitted over the 3 to 10 °C temperature range for each experiment. A polynomial fit, as with
equation (2), could not be obtained and instead a linear fit provided the optimal fit given the small number of data points obtained with a 13 min scanning time necessary for each size distribution. The slope of the linear fit between modal concentration and temperature gives the relative increase of each modal concentration per SST degree, relative to its 8-10 °C modal concentration. The linear fit was performed for each mode and each daily temperature experiment. Statistics for all experiments are shown Figure 6.

We observe that sea-spray concentrations in all modal sizes increased with decreasing temperature, but the sea-spray concentrations in the nucleation mode on average show the largest relative increase, followed by concentrations in the Aitken mode. Small particles are therefore most sensitive to temperature of the ocean surface water. Christiansen et al. (2019) also observed that the relative amount of small (<40 nm) particles increased at the coldest temperatures using synthetic seawater,
indicating that this is at least partly due to physical parameters. However, contrary to our observations they observed that larger (>300 nm) particles increased at the lowest temperatures. Using natural seawaters from a fjord in the Arctic, Zabori et al. (2012) found that the ratio of smaller (180 nm) to larger (570 nm) sea-spray particles increased at seawater temperatures colder than 3°C relative to temperatures above 3°C, which is consistent with our observations.

When a film breaks into a higher number of film drops, those are smaller (Lhuissier and Villermaux 2012; Poulain and Bourouiba, 2018), and therefore a decrease in size is expected from the higher numbers of SSA generated at colder temperatures. Additionally, when the main bubble bursts at the air-sea interface, numerous daughter bubbles of smaller diameter form at the edge of the main bubble (Bird et al. 2010; Millet et al. 2021), which may be the origin of additional drops of smaller sizes ejected to the atmosphere. The formation of daughter bubbles increases with the ratio of density to viscosity,
and further with the ratio of viscosity to surface tension. All of these variables increase with decreasing temperature, but it is difficult to quantify how their ratio evolves in the presence of unknown active chemicals.

## 4 Concluding remarks

We have observed a substantial increase in SSA flux as seawater temperature decreases. This finding aligns with previous observations from laboratory-based experiments using synthetic and natural seawaters (Hultin et al. 2011; Zabori et al. 2012; Salter et al. 2014; Christiansen et al. 2019). However, it contradicts the seawater temperature dependence of SSA fluxes inferred from ambient concentrations (Jaegle et al. 2011; Grythe et al. 2014). The reported temperature dependences from ambient concentrations were performed using either optical depth measurements (Jaegle et al. 2011) or sodium containing particles (Grythe et al. 2014), mostly represented by large accumulation mode particles that do not necessarily follow the same trends as the SSA number concentrations, mostly represented by smaller particles. We do find that larger SSA particle fluxes are not as sensitive to temperature as smaller ones. Also, in ambient air, it is difficult to discriminate between primary sea-spray production and secondary aerosol formation that either forms new particles or grows pre-existing particles, and may also be dependent on temperature. Therefore, the apparent contradiction between laboratory-based and ambient air sea-spray fluxes studies may be due to these two types of studies addressing differing aerosol sizes and processing/mixing states.

Comparison with results obtained from synthetic seawater in the literature is not straightforward as different conclusions are obtained in different studies. For example, Forestieri et al. 2018 report a monotonic increase of sea spray with increasing temperature, while Christiansen et al. (2019) reports a minimum in the sea-spray flux in the range 6-10°C, with both studies using synthetic sea spray and plunging jet systems, over the same range of temperature. As noted by Forestieri et al. (2018), variability between repeated experiments in their study (and so between different studies) could result from trace impurities of surfactants in the commercial synthetic salt solutions. For example, Christiansen et al. (2019) report a baseline TOC content in SIGMA sea salt <0.003% by mass, which corresponds to a significant amount of organic carbon of around 1.2 mg L$^{-1}$ in a 35 g L$^{-1}$ that is of the same order of magnitude as the amount of TOC in rich frontal waters of the present study (Fig 2a). Moreover, our sample with the lowest temperature dependence (March 18th, high Chl-a and TOC frontal seawaters) compares well with the temperature dependence flux model from Salter et al. (2014) and with synthetic seawater reported by Christiansen et al. (2019), while our highest temperature dependence samples (March 23rd and 24th, low Chl-a and TOC subantarctic seawaters) compare well with the temperature dependence experiments performed by Zabori et al. (2012) using arctic seawaters of different origins (Figure 4c). This indicates that neither Chl-a nor TOC are a good proxy for predicting the sea-spray flux temperature dependence.

Nanophytoplankton abundance was a major determinant of sea-spray number fluxes for sizes larger than 100 nm at moderate temperatures (SELL21), which was attributed to fatty acid concentration and surface tension effects. This is further supported in the present study, although the relationship to nanophytoplankton is not as strong as in SELL21. Here we show that the seawater temperature effect is inversely dependent on another phytoplankton group, the genus *Synechococcus* spp, which, if not coincidential, could indicate a link to SOM release during decay. However, there is a taxonomic heterogeneity within the

425 *Synechococcus* genus and within the picophytoplankton in general. Our results might be specific to *Synechococcus* spp. population of the South-Western Pacific Ocean and further work is required to investigate if the present relationship is applicable at a larger regional scale. In addition, more work is required to investigate the process by which *Synechococcus spp.* are related to the release of organic matter that specifically influences SSA flux at low temperature.

The dependence of SSA fluxes on water temperature is highest for nucleation and Aitken mode particles, but remains high for SSA sizes of the 100-200 nm size, which dominate the total SSA flux and are active as CCN. As a result, this phenomenon is expected to impact the seasonal fluctuations of sea-spray particles originating from cold surface waters, particularly in regions like the Southern Ocean. It may be also be relevant with respect to the cool-skin effect at the ocean-atmosphere interface. Average surface cooling by 0.2 K at global mean wind regimes (6 m sec$^{-1}$) relative to depths >1cm (Donlon et al., 2002) may

occur and may be greater under lower wind regimes or at night (Donlon et al., 2002, Marmarino and Smith, 2006). As SSA fluxes are generated at this interface they are influenced by the biophysical conditions encountered i.e. often characterized by an enrichment in biosurfactants (Wurl et al., 2011).

The IPCC report estimates an increase in average ocean surface temperature by 2.5 ° C by 2100, with consequences for marine biology (Bindoff et al. 2019) and future projections for the South-West Pacific also indicate a +2.5oC increase in surface

temperature by the end of the century with consequences for marine microbes and biogeochemistry (Law et al, 2018). Ocean warming is expected to expand the distribution range and cell abundance of *Synechococcus* (Morán et al., 2010; Flombaum et al., 2013). Based on our findings, it appears that higher seawater temperatures and increased abundance of Synechococcus would result in reduced sea spray fluxes at low temperatures. The combined effect of these factors could be additive or even synergistic, potentially amplifying the impact compared to each individual effect alone. However, higher temperatures and an

increased release of labile SOM could cause an effect on bacterial metabolism (Piontek et al., 2015)  that potentially favours the secretion of biosurfactants by heterotrophic bacteria. Potential changes in the abundance of *Synechococcus spp.* in response to temperature changes associated with climate change, and the resulting impact on CCN fluxes to the atmosphere and cloud formation should be investigated using regional models run under future climate conditions, to account for other climate-sensitive factors that influence sea spray fluxes. The quantification of process-based relationships between seawater

biogeochemistry and sea-spray cloud forming properties derived from the present study should enable improved actual simulations of cloud formation over the oceans of the Southern Hemisphere.

**Author contribution**

KSe and CL designed the Sea2Cloud voyage. KSe designed the experiments with the contribution of NB and MH and KSe and JT carried them out. TB, AC, KSa, SD and WD sampled seawater for biogeochemical analysis and AC, KSa, SD and KT

analysed sampled. AE supervised the work of TB. EF and CR contributed to aerosol measurement analysis. KSe prepared the manuscript with contributions from all co-authors.

**Aknowledgements**

We acknowledge the support and expertise of the Officers and Crew of the R/V Tangaroa. These results are part of a project that received funding from the European Research Council (ERC) under the European Union's Horizon 2020 research and innovation program (Grant agreement No. 771369), and was also supported by New Zealand SSIF funding to NIWA in the Ocean-Climate Interactions, and Flows and Productivity, Programmes. The Sea2Cloud project is endorsed by SOLAS.

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

| Mode | 7-9 °C | | | 2-3°C | | |
|---|---|---|---|---|---|---|
| | Dp (nm) | N | σ | Dp | N | σ |
| Nucleation | 10 | 0,006 | 1,5 | 8 | 0,012 | 1,5 |
| Aitken | 38 | 0,013 | 1,6 | 32 | 0,019 | 1,7 |
| Acc1 | 117 | 0,027 | 1,6 | 108 | 0,027 | 1,7 |
| Acc2 | 290 | 0,013 | 1,5 | 275 | 0,0115 | 1,5 |
| Coarse1 | 1000 | 0,0095 | 1,4 | 850 | 0,0065 | 1,4 |
| Coarse 2 | 2150 | 0,006 | 1,4 | 1800 | 0,003 | 1,4 |

**Table 1: Geometric mean diameter (Dp), normalized number concentration (N) and standard deviation (σ) of the 6 modes composing the median number size distribution at moderate (7-9°C) and low (2-3°C) SST.**

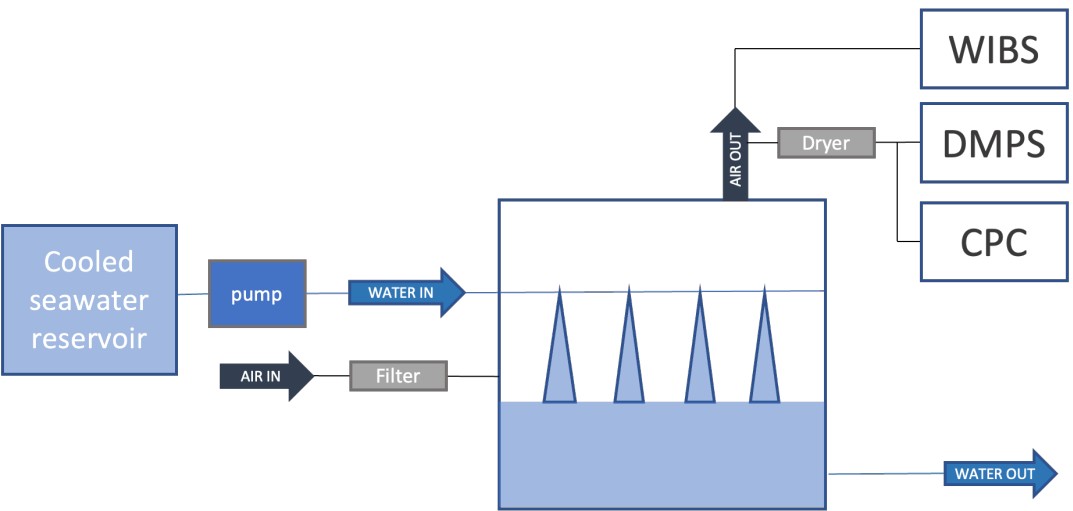

 **Figure 1: Schematics of the experimental set-up used to characterize the temperature dependence of SSA fluxes**

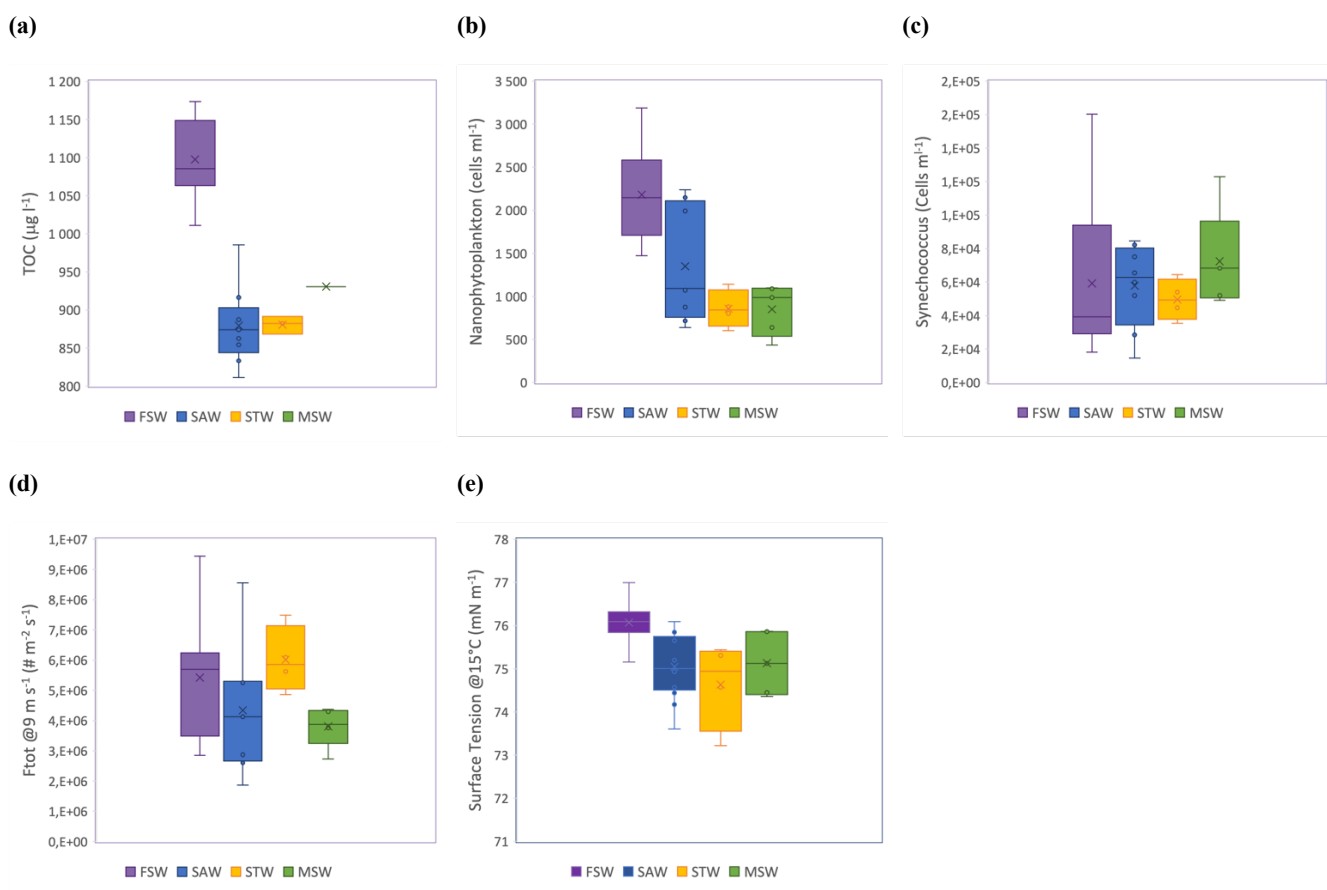

**Figure 2: Box-plot for the different seawater types of (a) TOC content, (b) nanophytoplankton (c) Synechococcus cell abundances**
 **and (d) surface tension of seawater samples for SST of 15 °C (right axis), and e) Total SSA flux measured at ambient STT (ranging 13-15°C) and normalized for 9 m s⁻¹ (left axis). Colour-shaded areas represent the different water masses, as indicated by the labels in (a), sampled during the Sea2Cloud voyage (Sellegri et al. 2023).**

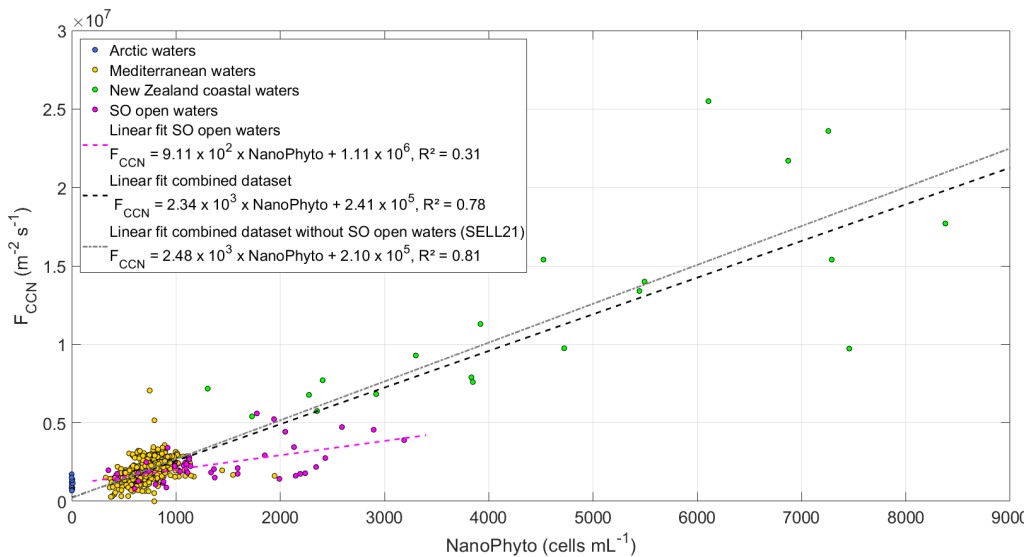

**Figure 3: SSA-derived CCN number fluxes as a function of NanoPhyto (Nanophytoplankton cell abundance) for the four regional datasets (Sea2Cloud data set indicated as SO for Southern Ocean), normalized to an equivalent wind speed of 9 m s$^{-1}$ and SST of 15°C. The linear fit to the Sea2Cloud data and combined dataset are shown with their corresponding equations and R², with the previously reported SELL21 relationship shown for comparison.**

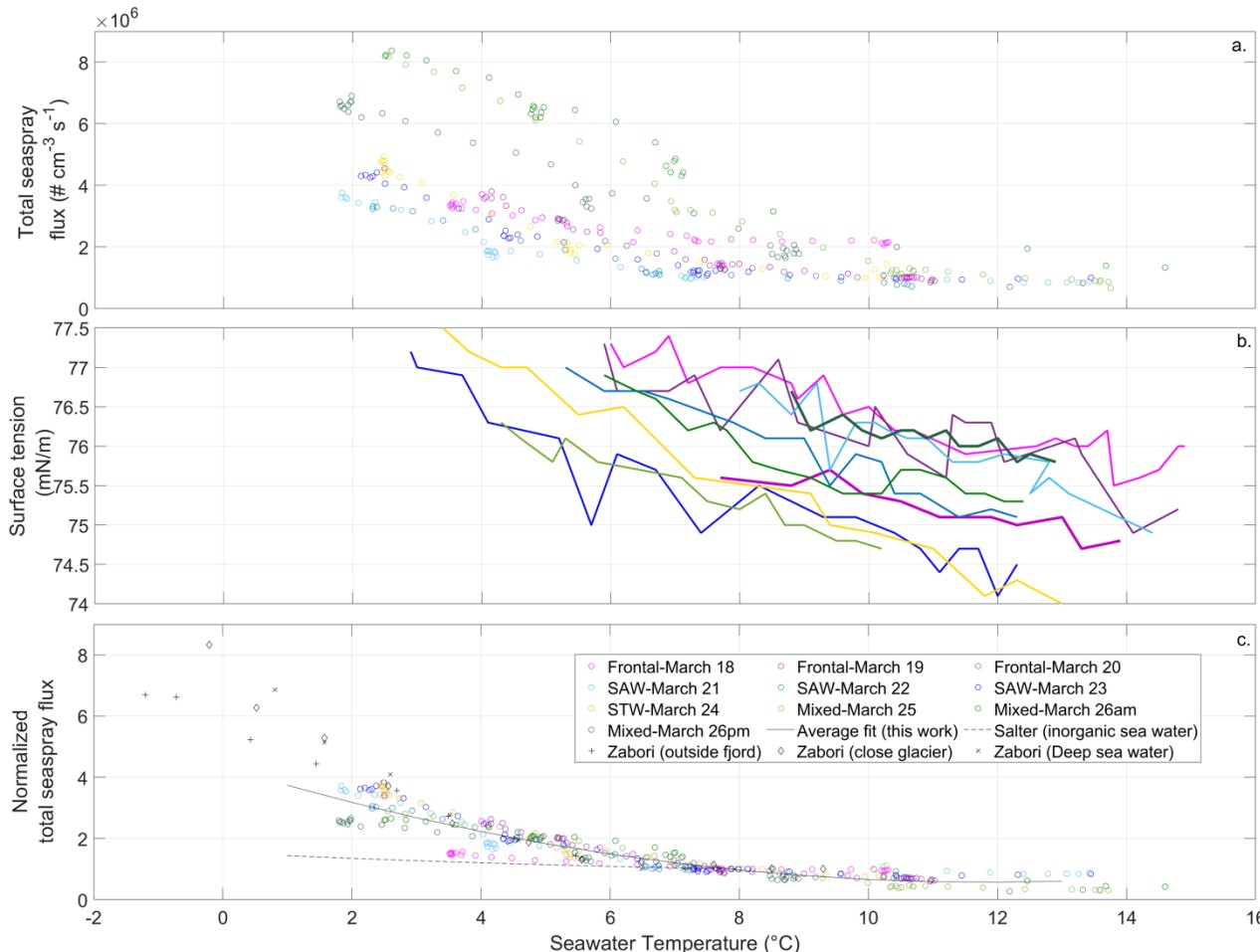

**Figure 4: (a) Sea-spray number flux calculated using eq. 1, as a function of seawater temperature during experiments performed in different water types. Colour code corresponds to water type and dates of sampling as in Figure 1. (b) Surface tension of the surface seawater sampled at 12:00 LT for different dates and seawater types, measured as a function of seawater temperature. (c) Sea-spray**
**number flux normalized to sea-spray flux measured at 8°C ($F_T/F_8$) as a function of seawater temperature at different dates and comparison to normalized fluxes ($F_T/F_8$) reported in Salter et al. (2014) for inorganic seawater and Zabori et al. (2012) for natural arctic seawaters.**

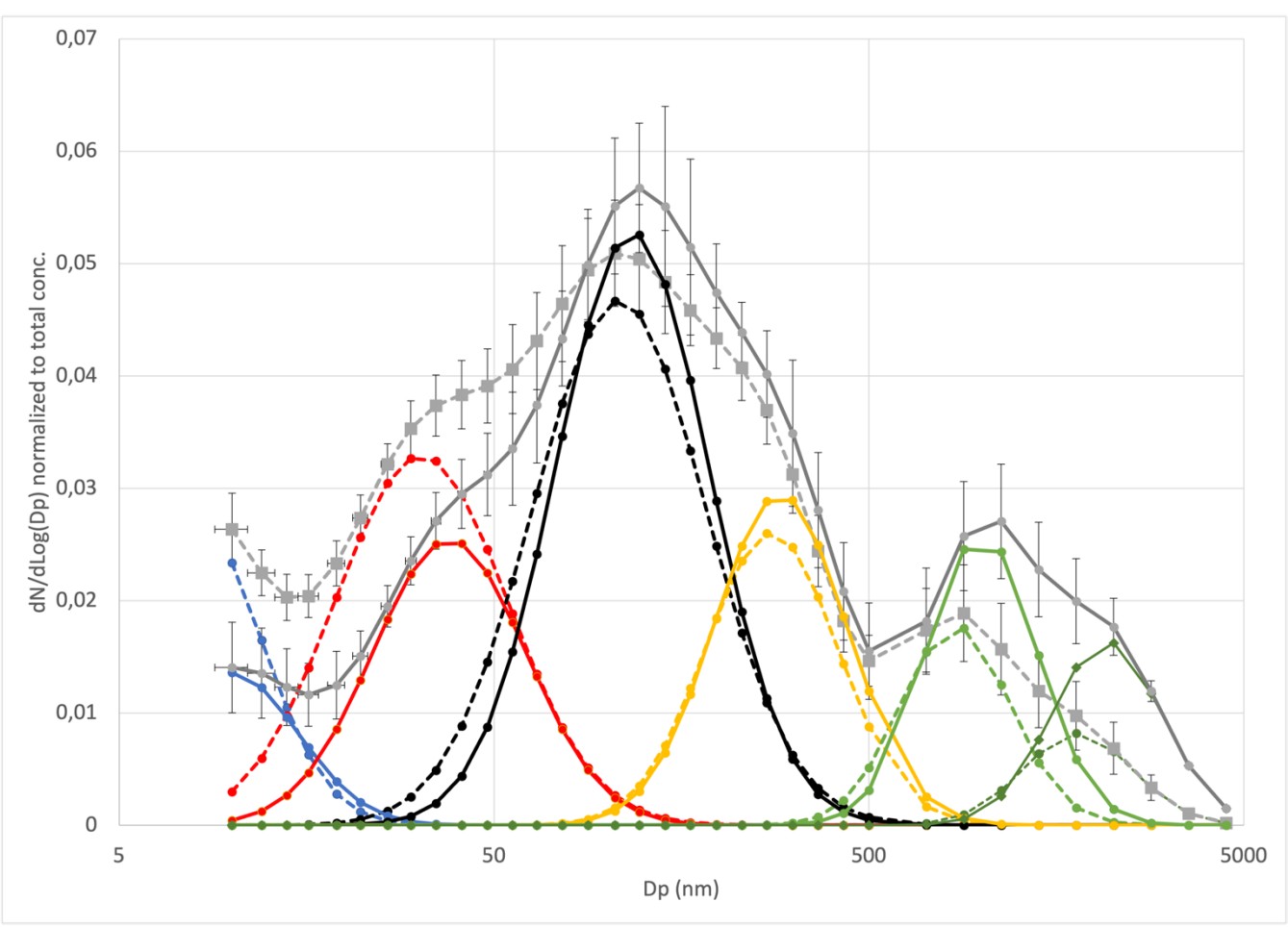

**Figure 5: Average sea-spray size distributions normalized to the total sea-spray concentrations for the temperature range 7 - 9 °C (plain lines) and 2-3 °C (dash lines) reconstructed from their decomposition in a combination of single log-normal (nucleation mode (blue), Aitken mode (red), first accumulation mode (black), second accumulation mode (yellow) and the two coarse modes (light and dark green). Error bars are from standard deviation on the averaged measured size distributions.**

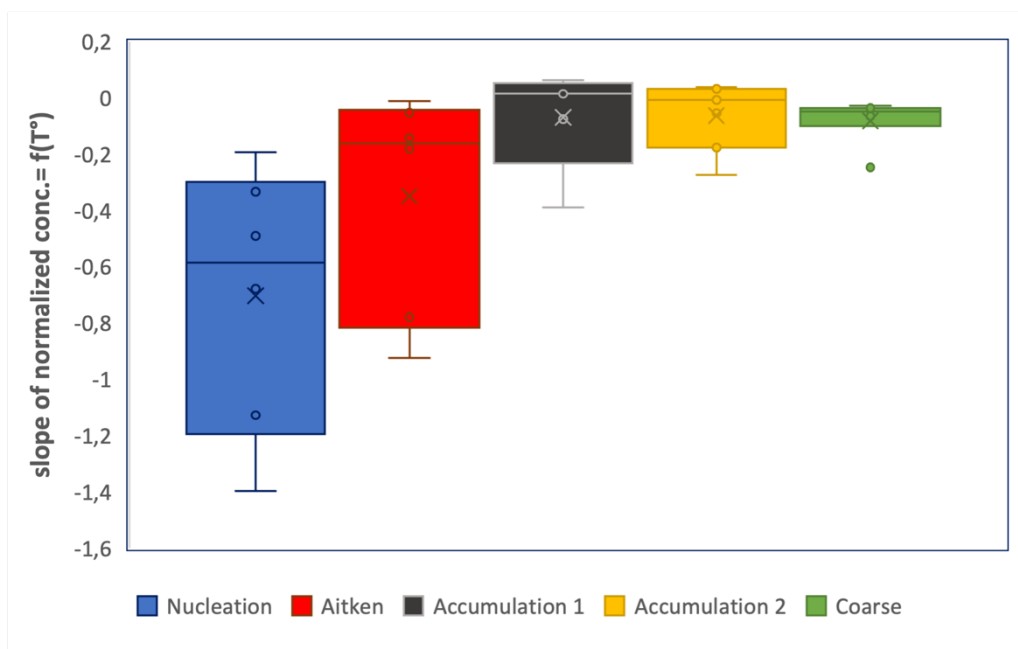

**Figure 6: Boxplot illustrating the dependence of sea spray fluxes to temperature for segregated sea spray size ranges: first, for each size range the ratios (sea-spray concentrations @T)/(sea spray concentration @ 8-10°C) were plotted as a function of seawater temperature for each of the 10 experiments, then slopes were derived from the linear fitting between these two variables and finally statistics were performed on the slopes obtained.**
