# Peer review of "Quantified effect of seawater biogeochemistry on the temperature dependence of sea spray aerosol fluxes"

_Atmospheric Chemistry and Physics, 2022_

## Author Comment (AC1)

**Answers to Reviewer1's comments**

We thank reviewer 1 for his/her constructive comments that we feel we have well addressed. Reviewer's comments are in black while our answers are in blue.

This paper presents measurements of sea spray aerosol particle number size distributions and calculated fluxes for particles produced from four different water masses during one sampling period. Surface tension measurements were done on water samples at different temperatures, and relationships between sea spray flux and water temperature are presented for the different water masses. This work is put in the context of previous work and relates the nanophytoplankton numbers to cloud condensation nuclei as well. Overall, this paper is interesting and relevant. Understanding sea spray production flux as a function of water temperature will be a good contribution to global modeling. There are some areas of the paper that can be improved, mostly through clarification of the methods and presentation of the results. Some general and specific comments are noted below.

General

The abstract states that future changes in seawater temperature will be important with climate change. In that case, why were such low temperatures chosen for the experiments? 2°C is outside of the range of the ambient temperatures measured (13-15°C). It would be helpful to have a comment on the global distribution of current sea surface temperatures and how they may change. For example, what fraction of the world's ocean has a sea surface temperature at 2°C? How will that change in the future?

The temperature range investigated is representative of a large fraction of the global oceans and in particular temperatures of 2°C and lower are observed at latitudes higher than 45° for both poles (Bopp et al. 2013), exactly where modelling exercises struggle to represent cloud occurrences and properties.
We now specify this in the introduction:
*"We investigate the temperature dependence of sea-spray aerosols generated from natural seawater of contrasting water masses of the South-Western Pacific Ocean over a temperature gradient equivalent to the 25-yr average summer seawater temperature range of the Southern Ocean (Auger et al. 2021) and relate this to the biogeochemical properties of the surface water"*

How sea surface temperature may change was mentioned in the conclusion:
*"The IPCC report estimates an increase in average ocean surface temperature by 2.5 ° C by 2100, with consequences for marine biology (Bindoff et al. 2019) and future projections for the South-West Pacific also indicate a +2.5ºC increase in surface temperature by the end of the century with consequences for marine microbes and biogeochemistry (Law et al, 2018)."*

Is it physically relevant to cool these specific water masses to 2°C? The statement made in the Conclusions is not fully supported. This study cooled ambient water, and many other factors will change with a warming ocean. It would have been more relevant to warm up the ambient water or present a larger range of temperatures.

The aim of our experiments is the study of the dependence of SSA fluxes on biologically produced chemicals that have differing physical macroproperties (surface tension, density, viscosity) at different temperatures. Temperature gradients applied over an hour provide a snapshot of the physical dependance of fluxes on these variables, the hypothesis being that biology does not change within this hour.

We agree with the reviewer that studying SSA flux temperature dependence is useful at higher temperatures for predicting atmospheric aerosol concentration over oceans in the future. We also acknowledge that these relationships may be less applicable under a future climate because of many other factors, including change or acclimatation of microbial groups to new temperature ranges. This is exactly what we discuss in the conclusion: how biology may change in a future climate, which *in turn*, will impact the spatial and seasonal temperature-related variability of SSA fluxes. Note that we provide a temperature dependent parameterization of SSA fluxes *as a function of the phytoplankton functional group*. Also note our conclusion that the complex question of the impact of climate change on SSA fluxes can only be addressed using Earth System Models.

*"Potential changes in the abundance of Synechococcus spp. in response to temperature changes associated with climate change, and the resulting impact on CCN fluxes to the atmosphere and cloud formation should be investigated using regional models run under future climate conditions, to account for other climate-sensitive factors that influence sea spray fluxes."*

The Methods should contain more details on the experiments and calculations, as mentioned below in the Specific Comments. There are some pieces in the Results and Discussion that should be moved into the Methods (i.e., the calculation of FCCN). Additionally, while references are included, there are some places that could use more explanation in the calculations. It also seems unnecessary to use "CCN" when it is simply defined as all particles over 100 nm. That could just be stated as its own variable.

We follow all suggestions here and detail changes we made in answer to specific comments below

Could you add some comments on how the temperature of the air would affect the particle number flux? The air temperature may influence the lifetime of the bubble at the surface. Would a large gradient in the temperature from the surface of the seawater to the air change the lifetime of the bubbles at the sea surface? What air temperature were these experiments done at? Was the air temperature in the headspace held constant, or was the whole system cooled?

Air temperature relative to the seawater temperature is indeed important for bubble lifetime, particularly via it's relation to the air RH, which impacts bubble film evaporation rate and therefore bubble lifetime. Our experiments were performed with ambient air temperature and therefore relatively constant air temperature and RH. Hence there is likely a change in the evaporation rate as the SST is decreased.

We have added a discussion on impact of RH in the main text, section 3.3.

"*Our experiments were performed with ambient air temperature and therefore relatively constant air temperature and RH of the incoming flushing air. Hence there is likely a change in the evaporation rate of the bubble film when the SST is decreased. However, the lower the SST in comparison to the air temperature above, the lower the evaporation of the film would be. Moreover, film thinning due to evaporation becomes more important relative to film thinning due to drainage only for very thin films (Miguet et al. 2021). In our system it is likely that bubbles films are broken by external forces before they reach these very thin films at which evaporation matters (below 1 micrometer for millimetric bubbles, achieved after a lifetime of several 10s of seconds, Miguet et al. 2021).*"

More text needs to be included discussing role of different factors on the surface tensions measured. The temperature at the time of measurement affects the surface tension, as stated. Additionally, salt concentrations in the water can change the surface tension as well. Because these are different water masses, it is likely that their salinities also change. Reporting a total surface tension might not be relevant, unless it is in the context of these changes in temperature and salinity.

We now address this in the revised text:
"*Given that both salinity and temperature influence surface tension, we performed a sensitivity test on the potential impact of salinity on the differences in surface tension observed for different seawater types. Salinity ranges of the different seawater types were 34.2-34.4 g $L^{-1}$ in SAW, 34.4-34.8 g $L^{-1}$ in frontal and mixed seawaters, 34.8-35.3 g $L^{-1}$ in STW and 34.4-34.8 g $L^{-1}$ in mixed seawaters. We calculate that these salinity ranges correspond to to ideal surface tension ranges of seawater at 15°C (Nayar et al. 2014) of 74.500-74.502 nN $m^{-1}$ in SAW, 74.502-74.514 nN $m^{-1}$ in frontal and mixed seawaters and 74.514-74.523 nN $m^{-1}$ in STW. Consequently, there is negligeable impact on surface tension within the range of salinities observed.*"

The discussion of the different biological factors influencing the particle flux at lower temperatures needs to be clarified. These are not necessarily species that would live at these temperatures. And it seems like any surfactants that would be in the water would have already been emitted at the ambient temperature. Is it possible that these species could die at these temperatures and then emit more surfactants or organics? Some more discussion of this would be useful.

Synechococcus spp. occur at a wide temperature range from 0 to 30°C but favour conditions around 10°C on a global scale (Flombaum et al., 2013, doi:10.1073/pnas.1307701110). While low temperature can induce stress in Synechococcus spp. acclimated at higher temperatures, differences occurring in metabolite production would be expected over the course of several hours (therefore over longer times than those of our experiments), while lowered temperatures are hypothesized to immediately slow down metabolic rates (Guyet et al., 2020, doi: doi.org/10.3389/fmicb.2020.01707). We therefore expect a relatively stable concentration and composition of organic matter and cell abundance during the application

of the temperature ramp and SSA flux measurements. As a consequence, the effect of lowered temperature on SSA fluxes is due to a physical impact of temperature on the bubble bursting mechanism in a solution that has the same biological and chemical composition.

We now include this discussion section 3.3

Towards the end of the paper, it is stated that the bubble films are more stable at colder temperatures due to the stabilization from surfactants. Why would there be more stabilization of surfactants at colder temperatures? This implies there are more or different surfactants when the temperature is changed.

Thanks for pointing this inconsistency. Different surfactants do not have different sensitivities to temperature, as now pointed out in answer to the previous comment. We added a discussion in section 3.3

*"The slope of surface tension to temperature does not differ from one sample to the other. This is expected as the Eötvös' equation states that the temperature dependance of the surface tension is the same for almost all liquids. SST-dependance of the evaporation rates should also be the same for all samples. One relevant variable that has varied with temperature in relation to the chemical composition of the solution is viscosity. Viscosity sensitivity to temperature depends not only on the concentration of organics but also on the ionic strength of the solution (pH, salinity), and it also increases exponentially with decreasing temperature (Mallet et al. 2020). An increase of viscosity implies an increase of the characteristic viscous time which leads to the decrease of the bubble film thinning rate (drainage) (Miguet et al. 2021). Bubble average lifetimes were found to be very sensitive to viscosity, especially when impurities are present (Miguet et al. 2021). Therefore, observed differences in thermal behaviours between seawater types would possibly be explained by differences in the sensitivity of different organic's viscosity to temperature. "*

The Results and Discussion section is somewhat short and could use more discussion of what these results mean. Some sections as mentioned below, could use more support for the statements. The Conclusions section starts as more of a discussion that could be moved up to the Results and Discussion. It would be better to have the key findings summarized in the Conclusions, and it should be clear what was measured in this study. The comparisons to the previous studies could be discussed in the previous section.

Thanks to the different comments received on this manuscript we significantly extended the Results and Discussion sections. We wish to leave the last section with some discussion, to put the main results together within a general framework. We renamed the section "concluding remarks" instead of conclusion.

Specific Comments

Line 105: Nothing to change. Just noting that March 2020 must have been an extra stressful time to be on a research cruise, and it is great that you were still able to finish the cruise and complete the experiments.

Thanks, we also consider ourselves lucky ☺

Line 121: FCCN is used in this sentence but has not yet been defined. Because it is defined later, it might be better to remove it from this sentence, or move the definition up. It seems like "F" is for "flux", but that should be stated explicitly here. And sub-scripting the "CCN" or not, should be consistent.
We now define and use FCN100 instead, as suggested.

Line 122: Please expand this sentence. It is unclear what two fluxes are being compared. It is interesting and relevant that the wind speeds are related to the air entrainment in the plunging jet system. It would be helpful to have the wind speeds and corresponding flow rates that were used written out here.

This is now described in the method section:
*"The flux of SSA was calculated from the SSA total number concentration, as follows:*

$$F_{tot} \ (\# \ m^{-2}s^{-1}) = \frac{CN_{tot}*Q_{flush}}{S_{tank}} \tag{1}$$

*where $CN_{tot}$ is the concentration of SSA measured from the MAGIC CPC, $Q_{flush}$ is the flushing air flowrate inside the tank's headspace, and $S_{tank}$ is the surface of seawater inside the tank.  In Sellegri et al. (2021), hereafter referred to as SELL21, the concentration of > 100 nm particles was used as a proxy for CCN concentration. For comparison to SELL21 we also calculated fluxes of SSA larger than 100  nm. The flux of CN100 ($F_{CN100}$) was calculated in a similar manner to Equation (1):*

$$F_{CN100} \ (\# \ m^{-2}s^{-1}) = \frac{CN_{100}*Q_{flush}}{S_{tank}} \tag{2}$$

*where $CN_{100}$ is the concentration of SSA with a diameter larger than 100 nm. Calibration experiments performed following the procedure of Salter et al. (2014), enabled to established that the air entrainment flowrate in our system is 4.5 Lair min-1 under the jet operational condition (seawater flowrate of 1.25 L min-1, orifices' diameters, jet distance to seawater surface). According to Long et al. (2011), the flux of air entrained (Fent) during wave breaking can be related to a wind speed at 10 m (U10) following:*

$$F_{ent} = 2*10^{-8}U_{10}^{3,74} \tag{3}$$

*Given that, we calculate that our plunging jet system simulated a bubble volume distribution equivalent to that produced at a wind speed of 9 m s-1.  For the data acquired with a seawater flowrate that deviated from 1.25 L min-1, fluxes were normalized to the 9 m s-1 equivalent windspeed with the following relationship:*

$$F_{normalized} = F_{original} * \frac{1.25^{2.4}}{Q_{SW}^{2.4}} \tag{4}$$

*Where Q_SW is the seawater flowrate. Equation (4) was obtained by varying QSW over a short period (less than an hour) and fitting the flux dependence to Q_SW. Normalization resulted in less than 30% change in the fluxes for 80% of the data."*

Line 127: It would be helpful to have more details on the surface tension measurements. Were these all done on board the ship, with fresh seawater, directly after sampling? How did the ship movement contribute to any uncertainties in these measurements? What volume of sample was collected for the temperature gradient experiments? Were these mixed to ensure a constant temperature throughout the sample?

Volume of samples were described in the text, and we now specify that the surface tension measurements were performed on board the ship directly after sampling:

*"The temperature gradient for surface tension measurements was achieved on board the ship on fresh seawater samples by first freezing 25 ml seawater sampled in Falcon tubes, with surface tension measured while the sample slowly warmed to ambient temperature; this took less than one hour which limited the time for any seawater biogeochemistry changes to occur."*

For this small volume there is no need for mixing as the temperature is homogeneous in the small glass bucket used for the analysis. The vibration of the ship may have influenced surface tension measurements, leading to the spread in measurements observed in Figure 4, however the uncertainty on measurements due to this effect is difficult to assess. We do not use surface tension measurement for their absolute value, but rather interpreted the sensitivity of these measurements to temperature and differences between seawater types qualitatively.

Line 132: How were the samples in the 10L carboys stored to prevent changes in the chemistry and biology? Please add more details on the aliquots and their storage prior to analyses. Were these analyzed on board the ship or later in the laboratory?

Sample bottles were either processed immediately or stored in the dark in ENGEL portable fridge/freezers units at in situ temperature for the water mass (max. operating temperature: 9°C) and processed within 8 hrs of collection. Sample volumes for filtering were determined from the Ecotriplet fluorescence data noted during sample collection. All biogeochemical analysis were performed on land post-voyage.

This is now specified in the method section.

Line 166: The paragraph prior to this line could be moved into the Methods section. This is mainly describing the water masses that were sampled and their dates. Starting at Line 166, there are some results from the measurements.

We followed the reviewer suggestion

Line 176: Add more explanation on how all of the particles greater than 100 nm can be considered CCN at 0.2% supersaturation.

We added more explanation in the method section and now refer to this section line 176. Note that there was a typing mistake and N100 correspond to CCN number concentrations at 0.1% supersaturation.

Figure 1 should be improved. The y-axes labels all appear to be stretched. The panels should be merged together into one figure, with one common x-axis. The shading and labels are nice to have. It would be helpful if this figure also had sea surface temperature and salinity. Maybe b and c could be combined, and SST could be added to a on the right axis. In d, why are there more surface tension markers than flux markers? Because the flux was calculated from the size distributions which were measured continuously, it seems like those could be at higher resolution or the same resolution as the surface tension. Error bars should also be added to these markers, especially the flux and the surface tension, to see any overlap in variability. (Also, change "STT" to "SST" in caption.)

Figure 1 was changed so that the variables of interest of the different seawater types could be statistically compared to one another.

[Figure]

In Figure 2 it is not clear what are measurements from this study and those from previous studies. The text states that there are four other datasets, but I do not see markers for the Sea2Cloud data. "SO" needs to be defined as well. If that is the Sea2Cloud data, then I am

not sure what the fifth dataset is that is referenced in the text. Overall, the caption could be a little more descriptive.

It is now specified that SO corresponds to the Sea2Cloud data set in now Figure 3's caption.

Line 190: I think this equation and all of the description of the FCCN calculations should be moved up into the methods. It would be helpful to have just an FCCN equation as well.

This is done

Line 202: This should be Figure 3.

Yes, we corrected this

Figures 3, 4, and 5 could be combined into a multi-panel figure, since they all have the same x-axis. Figures 3 and 5 should be combined, since they both contain sea spray flux as a function of temperature. It would be easier to compare the figures if they were together.

We follow the reviewer's suggestion but kept former Figures 3 and 5 separate. Figures 3 and 5 are difficult to merge into one figure because they do not have the same units and if plotted with a secondary y axis they would overlap too much.

Line 266: Interesting result that there is a shift in the shape of the size distribution.

Figure 6 needs a legend to describe the different colors. Additionally, some of the marker colors do not match the line colors (i.e., black dashed line with orange x's). It would be useful to have the same color scheme as Figure 7, to be consistent.

Corrected

Line 283: Can you add a little more description on the calculations going into Figure 7. It seems like it would be interesting to compare the Dp values shown in Table 1 for both temperature ranges.

We added the following text to better explain how now Figure 6 was done:

"*The slope of the linear fit between modal concentration and temperature gives the relative increase of each modal concentration per SST degree, relative to its 8-10 °C modal concentration. The linear fit was performed for each mode and each daily temperature experiment. Statistics for all experiments are shown Figure 6*."

And comment Table 1:
"We observe an average 15%  decrease of the modal diameters at the low temperatures compared to moderate temperatures which is consistent for all modes (Table 1)."

Line 294: This explanation is not consistent. Why would there be higher concentrations of surfactant in colder waters that would further stabilize the bubble film? Surfactant

concentration is not the only thing contributing to the surface tension and thus bubble lifetime.

See our answer above . We copy it here again:

Thanks for pointing this inconsistency. Different surfactants do not have different sensitivities to temperature, as now pointed out in answer to the previous comment. We added a discussion in section 3.3

*"The slope of surface tension to temperature does not differ from one sample to the other. This is expected as the Eötvös' equation states that the temperature dependance of the surface tension is the same for almost all liquids. SST-dependance of the evaporation rates should also be the same for all samples. One relevant variable that has varied with temperature in relation to the chemical composition of the solution is viscosity. Viscosity sensitivity to temperature depends not only on the concentration of organics but also on the ionic strength of the solution (pH, salinity), and it also increases exponentially with decreasing temperature (Mallet et al. 2020). An increase of viscosity implies an increase of the characteristic viscous time which leads to the decrease of the bubble film thinning rate (drainage) (Miguet et al. 2021). Bubble average lifetimes were found to be very sensitive to viscosity, especially when impurities are present (Miguet et al. 2021). Therefore, observed differences in thermal behaviours between seawater types would possibly be explained by differences in the sensitivity of different organic's viscosity to temperature. "*

Line 348: More explanation is needed to make this claim. It is not entirely clear that the results of this study support the idea that with a warmer ocean, there will be less sea spray flux. The temperatures measured here were colder than ambient. In order to make this statement, it would have been better to warm the water instead. There are a lot of factors that will change in a warming climate, so this needs to be clarified.
Citation: https://doi.org/10.5194/acp-2022-790-RC1
See our answer above. We copy it here again:

The aim of our experiments is the study of the dependence of SSA fluxes on biologically produced chemicals that have differing physical macroproperties (surface tension, density, viscosity) at different temperatures. Temperature gradients applied over an hour provide a snapshot of the physical dependance of fluxes on these variables, the hypothesis being that biology does not change within this hour.

We agree with the reviewer that studying SSA flux temperature dependence is useful at higher temperatures for predicting atmospheric aerosol concentration over oceans in the future. We also acknowledge that these relationships may be less applicable under a future climate because of many other factors, including change or acclimatation of microbial groups to new temperature ranges. This is exactly what we discuss in the conclusion: how biology may change in a future climate, which *in turn*, will impact the spatial and seasonal temperature-related variability of SSA fluxes. Note that we provide a temperature dependent parameterization of SSA fluxes *as a function of the phytoplankton functional*

*group.* Also note our conclusion that the complex question of the impact of climate change on SSA fluxes can only be addressed using Earth System Models.

*"Potential changes in the abundance of Synechococcus spp. in response to temperature changes associated with climate change, and the resulting impact on CCN fluxes to the atmosphere and cloud formation should be investigated using regional models run under future climate conditions, to account for other climate-sensitive factors that influence sea spray fluxes."*

About warming instead of cooling:

The temperature range investigated is representative of a large fraction of the global oceans and in particular temperatures of 2°C and lower are observed at latitudes higher than 45° for both poles (Bopp et al. 2013), exactly where modelling exercises struggle to represent cloud occurrences and properties.

We now specify this in the introduction:

*"we investigate the temperature dependence of sea-spray aerosols generated from natural seawater of contrasting water masses of the South-Western Pacific Ocean over a temperature gradient equivalent to the 25-yr average summer seawater temperature range of the Southern Ocean (Auger et al. 2021) and relate this to the biogeochemical properties of the surface water"*

---

## Author Comment (AC2)

**Answers to Reviewer2's comments**

We thank reviewer 2 for his/her constructive comments that we feel we have well addressed. Reviewer's comments are in black while our answers are in blue.

In this manuscript, the authors conducted a study during a research cruise in the Southern Ocean using a sea spray simulation chamber to generate nascent sea spray aerosol. The authors varied the seawater temperature in the chamber while steaming across different water masses to investigate the impact of seawater temperature and the biogeochemical state of the ocean on the sea spray aerosol flux.

As the authors allude to in their introduction, this research question has received significant attention in the scientific community. Although it is increasingly evident that seawater temperature impacts sea spray aerosol flux, there are still differences in both the magnitude and direction of the relationship between seawater temperature and sea spray aerosol flux, which may be due to variations in experimental approaches and differences in water chemistry and biology across different studies.

Although the study did not resolve the longstanding issue of the differences in the relationship between seawater temperature and sea spray aerosol flux found in the literature, my major criticism of the work is not the lack of scientific significance. Rather, the quality of the manuscript is compromised due to a general lack of attention to detail. The methods section omits critical details, rendering it impossible to evaluate the authors' findings. Furthermore, the work is poorly presented, making it difficult to understand the authors' intended message. In addition, the study overlooks numerous uncertainties, making it impossible to verify some of the authors' claims. In summary, the study feels rushed and fails to do justice to the authors' effort in collecting the dataset.

Therefore, I strongly recommend that the authors revise the manuscript substantially and focus on presenting the methods and results more clearly. As a result, I am afraid that I can only recommend rejecting the article in its current form. Below I outline in more detail the major issues I have identified with the manuscript.

Methods were only briefly presented in the original manuscript because they were already extensively detailed in previous published works (Schwier et al. 2015, Schwier et al. 2017, Trueblood et al. 2021, Freney et al. 2021) and for N100 fluxes in Sellegri et al. 2021. The experiment is also described in Sellegri et al. 2023. This last reference is cited in the present paper and publicly available in free access to read (https://journals.ametsoc.org/view/journals/bams/aop/BAMS-D-21-0063.1/BAMS-D-21-0063.1.xml), in case the reviewer needed more details than provided in the present work.

1. Schwier, A. N., C. Rose, E. Asmi, A.M. Ebling, W.M. Landing, S. Marro, M.-L. Pedrotti, A. Sallon, F. Iuculano, S. Agusti, A. Tsiola, P. Pitta, J. Louis, C. Guieu, F. Gazeau, and **K. Sellegri**

Primary marine aerosol emissions from the Mediterranean Sea during pre-bloom and oligotrophic conditions: correlations to seawater chlorophyll-a from a mesocosm study, Atmo.Chem. Phys., 15, 7961-7976, doi:10.5194/acp-15-7961-2015, 2015

2. Schwier A.N. , **K. Sellegri**, S. Mas, B. Charrière, J. Pey, C. Rose, B. Temime-Roussel, D. Parin, J.-L. Jaffrezo, D. Picard, R. Sempéré, N. Marchand and B. D'Anna, "Primary marine aerosol physical and chemical emissions during a nutriment enrichment experiment in mesocosms of the Mediterranean Sea, Atmos. Chem. Phys., 17, 14645-14660, https://doi.org/10.5194/acp-17-14645-2017, 2017

3. **Sellegri Karine**, Alessia Nicosia, Evelyn Freney, Julia Uitz, Melilotus Thyssen, Gérald Grégori, Anja Engel, Birthe Zäncker, Nils Haëntjens, Sébastien Mas, David Picard, Alexia Saint-Macary, Maija Peltola, Clémence Rose, Jonathan Trueblood, Dominique Lefevre, Barbara D'Anna, Karine Desboeuf, Nicholas Meskhidze, Cécile Guieu and Cliff S. Law Surface ocean microbiota determine cloud precursors, *Sci Rep* **11,** 281 https://doi.org/10.1038/s41598-020-78097-5, 2021

4. Jonathan V. Trueblood, Alesia Nicosia, Anja Engel, Birthe Zäncker, Matteo Rinaldi, Evelyn Freney, Melilotus Thyssen, Ingrid Obernosterer, Julie Dinasquet, Franco Belosi, Antonio Tovar-Sánchez, Araceli Rodriguez-Romero, Gianni Santachiara, Cécile Guieu, and **Karine Sellegri**, A Two-Component Parameterization of Marine Ice Nucleating Particles Based on Seawater Biology and Sea Spray Aerosol Measurements in the Mediterranean Sea, Atmos. Chem. Phys., 21, 4659–4676, https://doi.org/10.5194/acp-21-4659-2021, 2021

5. Evelyn Freney, **Karine Sellegri**, Alessia Nicosia, Jonathan T. Trueblood, Matteo Rinaldi, Leah R. Williams, André S. H. Prévôt, Melilotus Thyssen, Gérald Grégori, Nils Haëntjens, Julie Dinasquet, Ingrid Obernosterer, France Van-Wambeke, Anja Engel, Birthe Zäncker, Karine Desboeufs, Eija Asmi, Hilka Timmonen, and Cécile Guieu, Mediterranean nascent sea spray organic aerosol and relationships with seawater biogeochemistry, Atmos. Chem. Phys., 21, 10625–10641, https://doi.org/10.5194/acp-21-10625-2021, 2021

Major issues

The authors must provide a clear description of the sea spray simulation system they used in the methods section, as it is increasingly evident from the literature that the scale of laboratory systems used to generate nascent sea spray impacts the relationship between the number and size of aerosols generated as seawater temperature changes.

If the reviewer has the knowledge of a study intercomparing plunging jet systems and concluding their size is a main factor influencing the shape of the SSA size distribution as a function of temperature, we would like to read and cite this. We believe that the main factors are the jet orifice size, the jet flowrate, and the distance to the seawater surface (these impact the amount of air entrained in the seawater) rather than size. Operating the system in a continuous or disruptive manner, and the distance between two jets is probably also influencing the way bubbles are prematurely broken or not. In our system bubbles did not have any contact with the tank's walls, and provide SSA size distributions very similar to those observed in the natural clean marine sector of the region investigated (Sellegri et al. 2023).

In the current version of the manuscript, the authors have only referred to another paper (Sellegri et al. 2022) which is not included in the reference list. Given this, I assume that the system used is the same as that described in Schwier et al. (2015), …

The paper Sellegri et al. 2022 took a very long time to be processed - it was supposed to be published and accessible before the present manuscript – however, it is now available as Sellegri et al. 2023. It is clearly stated in the original text that: "Sea spray was continuously generated with a plunging jet system, as described in detail in Sellegri et al. (2022) and previously used in Schwier et al. 2015 and 2017, Trueblood et al. 2021, Freney et al. 2021 and Sellegri et al. 2021.", with the last 4 references accessible in ACP. Sellegri et al. 2023 is in now accessible and included in the reference list.

… which is relatively small compared to other systems used for simulating sea spray aerosol generation in the laboratory.

Although our system is several times smaller than systems such as those described in Dall'Osto et al. (2022), this reduces the seawater residence time, so limiting changes in biology or sedimentation of large phytoplankton species that occur in larger chambers (Dall'Osto et al. 2022). In addition, a smaller system also eliminates potential gas-phase reactions with air that may occur in other larger systems. We now clarify these advantages in the Method section:

"*Given the jet flowrates of 1.2 LPM, the relatively small seawater volume results in low residence time (4 min), so preventing changes in biology or sedimentation of large species that occur in larger chambers (Dall'Osto et al. 2022). The small dimensions of our system also correspond to a short residence time of air in the headspace (12s), also preventing potential gas-phase reactions with lab air.*"

The water depth in the system used by Schwier et al. (2015) was only 10 cm deep, which may have resulted in increased interaction between bubbles and the chamber walls.

Our plunging jets are equally spaced along the chamber diagonal and penetrate the seawater volume at a depth of 7 cm, and therefore do not interact with the chamber walls. Now specified in the method section.

Moreover, the sea spray chamber used by Schwier et al. (2015) had multiple plunging jets, and there may be differences between this setup and those that use single plunging jets. However, since the details of the chamber used are not clear, it is currently impossible to determine this.

This was stated in the previous literature that we cited, but it is now specified that we use 8 jets. There are indeed likely differences with the multiple jet system used in different studies (including the MART), with a one jet system.

Regarding the air entrainment rates described by the authors, how they were obtained is unclear. As the authors normalized their fluxes to this parameter, a clear description of the methodology used to determine the air entrainment rates is necessary in the manuscript.

We measured the air entrainment in our system using an approach similar to Salter et al. 2014. The procedure to use this air entrainment flux to derive an equivalent wind speed is described in the text:

*Calibration experiments performed following the procedure of Salter et al. (2014), enabled to established that the air entrainment flowrate in our system is 4.5 Lair min-1 under the jet operational condition (seawater flowrate of 1.25 L min$^{-1}$, orifices' diameters, jet distance to seawater surface). According to Long et al. (2011), the flux of air entrained (Fent) during wave breaking can be related to a wind speed at 10 m (U10) following:*

$$F_{ent} = 2 * 10^{-8} U_{10}^{3,74}$$ (3)

*Given that, we calculate that our plunging jet system simulated a bubble volume distribution equivalent to that produced at a wind speed of 9 m s$^{-1}$.*

 The authors need to provide a more detailed description of how they controlled the temperature of the sea spray chamber. While they mention the use of a 50 L temperature-controlled reservoir, it is unclear how this was connected to the sea spray chamber. Specifically, it is not clear whether the sea spray chamber was immersed in the reservoir or connected to it via some other means. A schematic of the experimental setup would be useful in clarifying this point. Further, how were the temperature experiments conducted?

A schematic of the experimental setup is provided in Sellegri et al. 2023, and is now also presented in Figure 1.

The authors mention that they applied temperature gradients ranging from 2°C to 15°C to the seawater over approximately 1 hour, but the exact form of these experiments is unclear. Given that these experiments are crucial to the study, it would be helpful for readers to see a typical experiment as a figure in either the main manuscript or the supplementary materials. This would provide more context and enable better understanding of the results.

All temperature experiments are reported Figure 4a (given that fluxes vary as SSA concentrations as stated equation 1).

In addition, these are fast temperature ramps which leads to the question of how repeatable the measurements were. Were any experiments conducted over a longer time period at constant temperature to determine the impact of quickly ramping the temperatures on the fluxes versus holding the system at a steady temperature?

Fast changes were applied so the seawater biology did not have the time to react to temperature changes as our goal was to investigate the physical dependance of fluxes to instant biogeochemistry. Holding the system for longer time periods would, in our opinion, lead to unrealistic changes in biology or chemistry and also our reservoir had a limited

volume that prevented these tests, so we did not conduct experiments at constant temperature. Now specified in the text.

There are some important details missing about the aerosol measurements. It is unclear how the instruments were connected to the sea spray chamber. Were all instruments connected through a single connection, and was the sampling conducted isokinetically?

The instrumental set-up is now shown in Figure 1

The type of differential mobility particle sizer used is also not specified. Was it purchased or custom-built in-house? Additionally, it is unclear whether an impactor was used to prevent particles larger than 500 nm or some other cutoff from entering the instrument. It is also unclear whether the aerosols were dried before sampling or whether they were measured under ambient conditions in the sea spray chamber, and if so, what was the relative humidity of the sample. Again, some of these details could be better explained with the inclusion of a schematic of the setup.

All of these details are given in Sellegri et al. 2023, now reported here again.

*" For submicron particles, SSA were taken through a ¼ inch stainless steel line to a 1-m long silica gel diffusion drier followed by an impactor with PM1 diameter cutoff. Particle size distributions were monitored by a differential mobility particle sizer system (DMPS) at 1 LPM, with a separate line to a condensation particle counter (MAGIC CPC, flowrate 0.3 LPM) connected in parallel for validation. The DMPS system was preceded by a soft X-ray aerosol neutralizer (TSI Model 3088) and consisted of a TSI-type custom-built differential mobility analyzer (length 44 cm) operated at a sheath flow rate of 5.0 L/min for selecting particle sizee range of 10-500 nm across 26 size bins during a 13 min 40s scan and a TSI CPC model 3010. Relative humidity at the inlet was monitored, and kept below 35% at all times. Another short, smooth curvature antistatic Teflon ½ inch line brought the generated SSA to a Waveband Integrated Bioaerosol Sensor (WIBS) for diameters ranging from 500 nm up to 4500 nm"*

In their study, the authors use the flux of particles larger than 100 nm as a proxy for cloud condensation nuclei (CCN). However, they should provide a detailed explanation of how they obtained this flux. Although they mention using the flush air flow and water surface of the tank, they fail to clearly explain the process. To help the reader understand, the authors should provide a mathematical explanation of the process and how they normalized it to wind speed. Moreover, the authors mention using air entrainment, but the details of how this was done are unclear, and it should be explained to the reader. To make it easier to understand, the authors should describe the process used mathematically.

Text has been modified:

*"The flux of SSA was calculated from the SSA total number concentration, as follows:*

$$F_{tot} \; (\# \, m^{-2} s^{-1}) = \frac{CN_{tot} * Q_{flush}}{S_{tank}} \qquad\qquad (1)$$

where $CN_{tot}$ is the concentration of SSA measured from the MAGIC CPC, $Q_{flush}$ is the flushing air flowrate inside the tank's headspace, and $S_{tank}$ is the surface of seawater inside the tank. In Sellegri et al. (2021), hereafter referred to as SELL21, the concentration of > 100 nm particles was used as a proxy for CCN concentration. For comparison to SELL21 we also calculated fluxes of SSA larger than 100 nm. The flux of CN100 ($F_{CN100}$) was calculated in a similar manner to Equation (1):

$$F_{CN100} \; (\# \, m^{-2} s^{-1}) = \frac{CN_{100} * Q_{flush}}{S_{tank}} \qquad\qquad (2)$$

where $CN_{100}$ is the concentration of SSA with a diameter larger than 100 nm. Calibration experiments performed following the procedure of Salter et al. (2014), enabled to established that the air entrainment flowrate in our system is 4.5 Lair min-1 under the jet operational condition (seawater flowrate of 1.25 L min-1, orifices' diameters, jet distance to seawater surface). According to Long et al. (2011), the flux of air entrained (Fent) during wave breaking can be related to a wind speed at 10 m (U10) following:

$$F_{ent} = 2 * 10^{-8} U_{10}^{3,74} \qquad\qquad (3)$$

Given that, we calculate that our plunging jet system simulated a bubble volume distribution equivalent to that produced at a wind speed of 9 m s-1. For the data acquired with a seawater flowrate that deviated from 1.25 L min-1, fluxes were normalized to the 9 m s-1 equivalent windspeed with the following relationship:

$$F_{normalized} = F_{original} * \frac{1.25^{2.4}}{Q_{SW}^{2.4}} \qquad\qquad (4)$$

Where $Q_{SW}$ is the seawater flowrate. Equation (4) was obtained by varying QSW over a short period (less than an hour) and fitting the flux dependence to $Q_{SW}$. Normalization resulted in less than 30% change in the fluxes for 80% of the data."

The authors conducted an experiment to measure the surface tension of seawater at different temperatures. They froze the water samples and then allowed them to warm up to 15°C over approximately one hour. It would be helpful to include a typical experiment's data plotted in the supplement or manuscript (this maybe figure 4 but it is not completely clear).

Yes, this was Figure 4 (which is now Figure 4b). We make this very clear now in the method section.

"Results of surface tension measurements as a function of sample temperature during unfreezing are shown in Figure 4b"

The authors suggest that the short time period of the experiment reduced the impact of changing biogeochemistry on their measurements. However, they do not discuss the potential impact of freezing on the surface tension. For instance, freezing could rupture phytoplankton cells present in the sample, releasing organic matter into the water, which could impact the surface tension. Did the authors conduct an experiment where they

measured the surface tension of a fresh sample at ambient temperature, froze it, and then returned it to the same temperature? If so, the measurements should be similar if freezing had limited impact, and this potential issue should be discussed.

Yes, freezing may have impacted surface tension measurements, as it does for many offline analysis when samples need to be taken back from ships to the laboratory. We did not conduct the tests suggested by the reviewer, and now warn the reader that a bias in the surface tension may have occur due to the impact of freezing. This effect should be investigated separately, as we anticipate that the impact is dependent on biology and would be a nice subject of investigation.

*"A bias may exist in the surface tension measured here after samples have been frozen, compared to the surface tension of a sample that would have not experienced freezing, due to the impact of freezing on, for example, the rupture of phytoplankton cells releasing organic matter. Future studies should investigate how freezing may impact surface tension."*

Furthermore, to rule out contamination from the sample tubes, were any measurements of pure water taken in the Falcon tubes? Including this information would enhance the clarity of the authors' findings.

Blank measurements were regularly performed using milliQ water, and we expect minimal contamination of Falcon tubes by surfactants.

The authors have presented time series of different parameters in Figure 1, but there are some significant issues with the data presentation. Firstly, the linear axis appears to have the same distance between the values of 0.00E+00 and 1.00E+06 and the values of 1.00E+06 and 2.00E+06.
Additionally, all the subscripts are missing from the axis labels and legend entries, which may seem minor, but it suggests carelessness on the part of the authors.
Moreover, the authors argue for a trend in this data, which is impossible to determine considering there are no error bars on the data points. Without an idea of the uncertainty on each data point, it is difficult to find a trend in the data when it is so noisy. Furthermore, it is not clear what the authors have plotted in Figure 1. Have they plotted the integrated number flux of all particles following normalization, or is it the integrated number flux of all particles larger than 100 nm? The authors should clarify this point to help readers better understand their results.

We modified Figure 1 (now Figure 2) and accounted for all the reviewers observations

The authors initially mentioned fitting "single lognormal modes" to their measurements of aerosol size distribution. However, in the following sentence, they reported finding "four modes in the submicron range and two modes in the supermicron range". This description is unclear and adds to the manuscript's overall confusion. Upon examining Figure 6, it becomes evident that the authors actually fitted a series of lognormal modes, not "single lognormal modes".

Now clarified in text:

*"These were averaged over two different temperatures ranges, 2-3 °C and 7-9°C, and fitted with a combination of single lognormal modes (Figure 5)."*

However, they did not provide any information about their fitting procedure or the quality of the fits. It is generally recommended to present data and its associated uncertainty (e.g., mean/median and standard deviation/error) before comparing contrasting datasets. If the authors wish to include fits, they could be presented in another panel or the supplement. Comparing the actual data across different temperatures is crucial to determining whether differences exist between the presented temperature regimes. Without the data, it is impossible to determine whether such differences exist.

A mode fitting procedure was applied for each experiment to the average size distribution measured in the coldest temperature range and also the highest temperature range, in order to visualize the change in the mean diameter of each mode, as reported Table 1 (and now commented in text). We now show the quality of the fitting procedure in Figures S1 and S2 for the respective temperature ranges. Also, standard deviations were added to Figure 6 (now figure 5) highlighting how significantly the two average size distributions were.

[Figure]

**Figure 5: Average sea-spray size distributions normalized to the total sea-spray concentrations for the temperature range 7 - 9 °C (plain lines) and 2-3 °C (dash lines) reconstructed from their decomposition in a combination of single log-normal (nucleation mode (blue), Aitken mode (red), first accumulation mode (black), second accumulation mode (yellow) and the two coarse modes (light and dark green). Error bars are from standard deviation on the averaged measured size distributions.**

[Figure]

**Figure S1: Measured sea-spray size distributions from merged SMPS and WIBS data, normalized to the total sea-spray concentrations and averaged for the temperature range 2-3 °C (dash grey line). Decomposition in a combination of single log-normal modes show the nucleation mode (blue), Aitken mode (red), first accumulation mode (black), second accumulation mode (yellow) and the two coarse modes (light and dark green) and reconstructed size distribution from the addition of individual modes (plain grey line). Error bars are from standard deviation on the averaged measured size distributions.**

[Figure]

**Figure S2: Same as S1 for the temperature range 7-9 °C.**

---

## Author Response (AR2)

Dear editor,
We believe we fully addressed the referees concerns as described below:

1) The presentation of some of the figures needs to be improved and the style of the figures should be consistent throughout the paper. Some are not as clear as most figures in this journal. For example: The new Figure 2 has no x-axes. Some of the ranges should be reduced to show the variability (i.e., TOC could range from 600 to 1300 instead). The axes labels should have units in a standard format (i.e., μg instead of microgramme written out; m-2 and s-1 need to be superscripted). There are grey boxes around some of the panels and the boxes don't seem to be centered on the graphs. Additionally, I think the individual markers need to be darker because they are hard to see on the boxes. It seems like Figures 2, 5, and 6 were made by a different program and have a different style than Figures 3 and 5. I think they should all have consistent formatting.

We reformatted Figure 2 as requested and in compliance with the formatting of the other figures.

2) Although the authors have included a schematic of the setup used for measurements, there are still important missing details. For instance, the dimensions of the tank are not provided. While the authors argue that the plunging jet does not interact with the tank's sidewalls and bottom, I am skeptical that the free-floating bubbles on the surface do not interact with the walls to some extent. It would be helpful if the authors could provide a photo of the system with the jet on to demonstrate the coverage of bubbles on the surface and their proximity to the walls before bursting. This information is also crucial for calculating fluxes from the chamber. According to the revised version, the authors assume that the entire water surface in the chamber generates particles, which contradicts their assertion that bubbles do not interact with the chamber walls. It is likely that less than 100% of the water surface acts as a particle source, and additional photos could assist the authors in estimating tank fluxes more accurately. In addition, it would be beneficial to provide information about the depth of the headspace, which would enable a comparison with the typical height reached by the jet and film droplets generated upon bubble bursting. This would help confirm whether only a few droplets were impacting the chamber roof, thereby affecting the measurement process.

We now provide the dimensions of as many items as we can of the system so the reviewer (and reader) can duplicate it. Of course free floating bubbles eventually meet the tank's walls, but the jet does not reach the tank's bottom (as we stated).

The text was modified as follows:

"Sea spray was continuously generated with a plunging jet system, as described in detail in Sellegri et al. (2023) and previously used in Schwier et al. 2015 and 2017, Trueblood et al. 2021, Freney et al. 2021 and Sellegri et al. 2021. The **10 L tank was operated with a 10 cm seawater depth, so jet and film drops did not interact with the tank's top locate 15 cm above the seawater level**. Given the jets total flowrate of 1.2 LPM, this relatively small seawater volume results in low residence time (4 min), so preventing changes in biology or

sedimentation of large species that occur in larger chambers (Dall'Osto et al. 2022). The small dimensions of our system also correspond to a short residence time of air in the headspace (12s),  preventing potential gas-phase reactions with lab air. **Eight plunging jets were created by flushing seawater through 1 micrometer orifices that were** equally spaced along **a ¼" stainless steal tube, located at 5 cm below the tank's top in the** chamber diagonal**. Jets** penetrate the seawater volume at a depth of 7 cm, and therefore do not interact with the chamber **bottom**. **Free floating bubbles could occasionally meet the tank's wall as they floated away from the center of the tank. For this reason and others such as the continuous jets vs intermittent wave breaking process, fluxes derived from our experiments, similarly to all controlled lab experiments, are necessarily different from the ones obtained from the natural wave breaking in the open ocean**. **Natural conditions were however mimicked as much as was possible."**

We do not believe providing a photo of the system would be of much value, as the bubble lifetimes (and so the "whitecap" surface coverage) do change significantly as a function of seawater biology and temperature, so a photo would be representative of only one experiment.
Moreover, the fraction of the tank's surface covered by bubbles is not taken into account in our flux calculation, neither it is assumed to be 100% . Instead, we use the flow of air entrained in the seawater, an easier parameter than the surface covered by bubbles, (related to whitecap coverage) to scale our flux to equivalent windspeed condition in the natural world. Indeed air entrainement flowrate is easier to accurately measure than whitecap coverage, and it can easily be prognosticated in models. The tank's surface is only used together with the flushing flowrate, to derive a number of sea spray particles produced across the seawater surface per unit of time, as a function of air entrained.

3) Similarly, there are still important details missing regarding the air entrainment measurements. Simply stating that the authors followed the same procedure as Salter et al. (2014) is insufficient. Upon reviewing that publication, it became clear that it did not provide enough information to replicate their work. For instance, the diameter of the tube used to enclose their plunging jet was not mentioned.
This is now mentioned

Additionally, while Salter et al. used a single plunging jet, the current study employed multiple parallel jets. It is crucial to clarify whether all the jets were enclosed simultaneously and if air entrainment was measured accordingly.
The air entrainment measurements experiment were performed on a single jet as Salter et al. (2014) did. Now specified.

The authors should provide details such as the diameter of the tube used, the depth to which it penetrated the water surface, the method used to measure airflow into the tube, and the number of measurements conducted (assuming an average was taken).

Text was added to describe the air entrainment measurement set-up:

"The set-up used to measure ($F_{ent}$) **reproduced one of the 8 plunging jets set in a separate, larger tank, with the same distance to seawater and seawater depth than the main experimental set-up. For the air entrainment measurements, the jet was enclosed in a ½" vertical plunging tubing (at 1 cm depth) connected to a TSI flowmeter. The seawater flowrate was varied from 150 to 400 ml min$^{-1}$ and the relationship between seawater flowrate and entrainment air flowrate was fitted to obtain a calibration curve of our set-up."** Air entrainment flowrate calibrations were performed at moderate temperatures around 20 °C **and also at lower temperatures that showed undetectable influence of the seawater temperature on the air entrainment flowrate**."

In addition, the authors state on line 75 that "The combination of viscosity, density and surface tension changes may also affect the volume of air entrained in the seawater and the total volume and number of bubbles formed." This implies that seawater temperature may impact air entrainment. As such, the authors should also state the water temperature that their air entrainment measurements were made at and how this impacts their flux calculations.

We actually made new air entrainment measurements experiments in which the seawater temperature was varied and did not detect any effect. Therefore we withdraw the above mentioned sentence "The combination of viscosity, density and surface tension changes may also affect the volume of air entrained in the seawater and the total volume and number of bubbles formed" and added the following text:
 "Air entrainment flowrate calibrations were performed at moderate temperatures close to 20 °C **and also at lower temperatures that showed undetectable influence of the seawater temperature (across the range 2°C-16°C) on the air entrainment flowrate**."

Minor comments.
In Figure 1a) It should be "microgram"
Done

In Figure 1e) The units are missing.
Done

Line 20 - This would read better as "We observed a significant increase in sea spray total concentration at temperatures below 8 °C. Specifically, at 2 °C, there was an average 4-fold increase compared to the initial concentration at ambient temperatures."
Done

Line 23 - Should read "Moreover, the temperature dependence varied based on the type of water mass and its biogeochemical properties."
Done

Line 25 - Would read better as "The temperature dependence of the sea spray flux was found to be inversely proportional to the abundance of the cyanobacterium Synechococcus in seawater. This relationship allows for parameterizing the temperature dependence of sea

spray emission fluxes based on Synechococcus, which can be utilized in future modeling exercises.

Done

Line 59 - Would read better as "Laboratory experiments using a plunging-jet sea spray generator provide a means to investigate the temperature dependence of sea-spray number flux (in contrast to sea spray mass flux) across various ranges of sea-spray size and temperature.

Done

Line 383 - Would read better as "We have observed a substantial increase in SSA flux as seawater temperature decreases. This finding aligns with previous observations from laboratory-based experiments using synthetic and natural seawaters (Hultin et al. 2011; Zabori et al. 2012; Salter et al. 2014; Christiansen et al. 2019). However, it contradicts the seawater temperature dependence of SSA fluxes inferred from ambient concentrations (Jaegle et al. 2011; Grythe et al. 2014).

Done

Line 409 - This sentence should be revised to make it clear that a previous campaign is being referred to ("SELL21").

We rephrased. We do refer to the present study.

"For example, Christiansen et al. (2019) report a baseline TOC content in SIGMA sea salt <0.003% by mass, which corresponds to a significant amount of organic carbon of around 1.2 mg L−1 in a 35 g L−1 that is of the same order of magnitude as the amount of TOC in rich frontal waters of the present study (Fig 2a)."

Line 420 - Would read better as "As a result, this phenomenon is expected to impact the seasonal fluctuations of sea-spray particles originating from cold surface waters, particularly in regions like the Southern Ocean.

Done

Line 431 - Would read better as "Based on our findings, it appears that higher seawater temperatures and increased abundance of Synechococcus would result in reduced sea spray fluxes at low temperatures. The combined effect of these factors could be additive or even synergistic, potentially amplifying the impact compared to each individual effect alone."

Done